Subject Areas:
behaviour/psychology/cognition

Keywords:
group hierarchy, leadership, political bias, trust game

Authors for correspondence:
Biljana Gjoneska
e-mail: biljanagjoneska@manu.edu.mk
Salvatore M. Aglioti
e-mail: salvatoremaria.aglioti@uniroma1.it

# Bound to the group and blinded by the leader: ideological leader–follower dynamics in a trust economic game

Biljana Gjoneska[1,2,3], Marco Tullio Liuzza[2,3,4], Giuseppina Porciello[2,3], Gian Vittorio Caprara[2] and Salvatore M. Aglioti[3,5]

[1]Academy of Sciences and Arts of North Macedonia, [2]Department of Psychology, 'Sapienza' University of Rome, Rome, Italy and [3]IRCCS Santa Lucia Foundation, Rome, Italy
[4]Department of Surgical and Medical Sciences, 'Magna Graecia' University of Catanzaro, Rome, Italy
[5]Department of Psychology, 'Sapienza' University of Rome and CLNS@SAPIENZA at the Italian Institute of Technology, Rome, Italy

BG, 0000-0003-1200-6672

Understanding the dynamics of trustworthiness in ideological contexts could influence human societies, affect electoral campaigns and ultimately impact democracy. We tested trust behaviour towards political leaders in a sample of 121 opposing/supporting voters assigned as trustors in an iterative trust game (TG). In two experiments, a famous Italian conservative leader (i.e. Silvio Berlusconi) or a famous non-politician were used as trustees in a predefined un/trustworthy TG, while trustors believed that mathematical algorithms reproduced trustee's real behaviour. Results revealed that depending on the group, voters either relied on the situation and adjusted to the behaviour of the out-group leader (in our case left-wing voters), or on their disposition for group-loyalty with respect for authority, thus failing to adjust to the behaviour of the in-group leader (in our case right-wing voters). Our findings suggest that: (i) complex voter–leader relations in politics are reflected in the simple trustor–trustee financial interactions from behavioural economics, and (ii) being bound to one's group and one's leader may affect the trust economic decisions of the followers.

# 1. Background

Trust (i.e. the willingness to expose vulnerability to a trustee based on positive expectations that it will not be misused for harmful purposes) and trustworthiness (i.e. perception of benevolence and reliability of a trustee) influence every social exchange, be it personal or professional, economical or political. People largely rely on trust when relating to others whether they are close friends, financial advisers or political leaders. Mounting behavioural and neural evidence indicates that trustworthiness evaluations of unfamiliar persons are made rapidly (in less than 100 ms) [1] and automatically [2]. However, situational and dispositional factors may affect trust and trustworthiness judgements, and impact basic social behaviours. Humans, like other social species, naturally tend to coalesce in groups along one or more dimensions (race, ethnicity, religion, sexual orientation and political affiliation) [3,4]. Grouping seems to imply that people are inclined to prefer the fellow members of their own group, not only when the group is meaningful but also within communities simply sharing arbitrary labels [5]. Political ideology, through cohesion of certain racial, ethnic and religious groups creates shared outlook, induces strong feelings of group-identification and represents a strong motif for group-differentiation [6,7].

Theories from political psychology (Moral Foundation Theory, Right-Wing Authoritarianism Theory and Social Identity Theory regarding political affiliation) indicate that group-related political bias may be characterized by the group-cohesive moral dimension and the authoritarian submission dimension [7–10]. While individualizing moral values (like fairness and reciprocity, with care for others and harm-sensitivity) are common for right and left ideologies, binding moral values (like in-group loyalty, respect for authority and concern for religious purity within one's own community) characterize specific partisanships, i.e. right-wing parties [8,9]. Furthermore, voters who score higher on binding moral values tend to support social attitudes that lead to in-group favouritism, with promotion of social hierarchy and inequality [11,12]. This, in turn, may lead to biased perception about trustworthiness of in-group members (especially if they are leaders placed high on the hierarchical ladder), and may produce bias in variety of social behaviours. The bias may also become evident on daily bases, and reflected in the daily decisions to trust like-minded partisans, not only on political issues but also on unrelated, non-political matters (the so-called epistemic spillover) [13]. For example, Marks et al. [13], found that the similarity with one's political views affects one's ability to make accurate assessment about their fellow's expertise in the domain of geometric shapes. Moreover, trusting in-group politicians (especially when they are powerful leaders) can happen even when this implies spreading disinformation [14].

It is also worth noting that in complex leaders–voters dynamic, the former are responsible for setting in motion political systems as services for the common good. Leaders use extended media presence in order to exert influence over potential voters, by means of projecting admirable personality traits, portraying them as desirable candidates [15,16]. Hence their roles (e.g. leaders as trustworthy and voters as trusting subjects) and the incentives (e.g. economic offers) may be reflected in the experimental design of the trust game (TG), a paradigm that has proved useful for the study of decision-making in an interactive fashion [17].

Although the TG has already been used to model political relations [18,19], or measure political bias [5,20–22], the trustees were either explicitly introduced as peers (through a short, written profile) [20] or implicitly inferred as such (by presenting unknown candidates for runners-up) [21], but never distinctly associated with the persona of a renowned political leader.

To the best of our knowledge, we are the first to explore the temporal unfolding of trust in a hierarchical voter–leader, iterated economic interaction. Specifically, we explored how the (un)trustworthy behaviour of a famous political leader shapes the behaviour of opposing (in our case LW) or supporting (in our case RW) group of voters. More specifically, we tested if and how the trust-behaviour of the voter depends on their own and leader's political orientation, as well as their mutual economic interaction. Note that in a standard TG an investor/trustor and a partner/trustee interact by exchanging money. The trustor is endowed with a fixed amount of money and informed about the opportunity to make an offer to the trustee. The money offered is then multiplied by the experimenter (usually by a factor of 3 or 4) and transferred to the trustee who has the opportunity to keep the multiplied amount of money or transfer back some of it, thus behaving fairly or not. We used a computerized TG, modified in accordance to Chang et al. [23] to test 58 right (RW) and 63 left (LW) politically oriented Italian voters. Participants believed that they played with an algorithm simulating either the behaviour of Silvio Berlusconi (SB; former Prime Minister of Italy and leader of the

event outline

| fixation | stimulus presentation | offer selection | additional information | pause | outcome revelation |

'your partner in this game will be Silvio Berlusconi'

1 to 10€

4 × offer

'please wait while the model computes Berlusconi's decision'

'Berlusconi has decided to reciprocate the offer. You gained _€'

trust condition: 80% reciprocation

distrust condition: 20% reciprocation

| 1000 ms | 1000 ms | untimed | 1000 ms | 4000 ms | 4000 ms |

timeline

**Figure 1.** Study-specific protocol of the trust game (TG). Each participant is randomly assigned to play one TG version (trust/distrust) with one character (control/experimental) and endowed with 10 euros that are 'renewed' on each of the 15 trials. Each trial begins with a fixation cross (1 s), followed by image-presentation of the character. The trustor's decision is marked with selection of the corresponding offer (untimed) and followed by calculation of the quadrupled investment (1 s). The trustee's decision is revealed after a pause (4 s) during which the supposed mathematical model computes trustee's behaviour. The trial ends with the final pay-off for the participant. Analogous images of trustees include RW leader Silvio Berlusconi (published by European People's Party) and TV host Piero Angela (published by Elena Torre), released under Creative Commons license, downloaded from Flickr and adjusted in GIMP by the authors.

centre-right coalition) or Piero Angela (PA; TV personality and writer of science-related popular books, i.e. a famous age-matched non-politician). In reality, participants were assigned to a version of the TG where the trustee could behave in a trustworthy (i.e. reciprocate 80% of the time) or untrustworthy (i.e. reciprocate 20% of the time) fashion. With the implementation of this experimental design we were able to capture the initial trust in two groups of participants towards an in-group versus out-group leader (hence detect potential positive versus negative bias towards him), and manipulate the leader's trustworthiness by obtaining a measure of change in participants' trust index. A schematic of the experimental task is provided in figure 1.

We hypothesized that the overall dynamics of trust would unfold in the following manner: in the supporting group of voters, we expected an in-group favouritism towards Berlusconi. Specifically, we presumed that it would be expressed at the beginning of the game (as higher amounts of initial investments), and persistent throughout the game in the trend of their investments (as a lack of adjustment to the trust/distrust condition) when they play with Berlusconi, when compared with Angela. Regarding the opposing group of voters, we expected out-group derogation towards Berlusconi, reflected in a lower amount of initial investments (as compared with Angela) that will also

persist throughout the game in the trend of their investments (as lack of adjustment to the trust/distrust condition). Alternatively, we expected an evidence-based behaviour correspondent with the condition of the game, irrespective of the partner, since none has the potential to instigate the in-group related favouritism.

# 2. Material and methods

## 2.1. General procedure

The study was carried out in accordance with the 1964 Helsinki Declaration. In addition, the study was approved by the Institutional Review Board at the Scientific Institute for Research Hospitalization and Health Care — Santa Lucia Foundation in Rome, Italy. Recruitment included: phone-mobilization of volunteers, on-site collection of students, and peer-to-peer nomination of classmates with a clear political interest and orientation. Selection concluded with an informed consent following a detailed account of the procedure by the experimenter, and signed assent by the admitted candidates. The study was conducted at the Department of Psychology, 'Sapienza' University of Rome. The laboratory was equipped with standard air-conditioning system (temperature maintenance) and illumination (lighting sustenance), chair and an LCD monitor (head positioning at 0.5 m approximate distance), E-Prime (Psychology Software Tools, Inc., Pittsburgh, PA, USA) for experiment creation and a PC workstation for experiment execution.

## 2.2. Participants

Selected candidates belonged to the same national (Italian) and ethnic (Caucasian) group. All of them were students at one of two universities, the public 'Sapienza', or the private Libera Università Internazionale degli Studi Sociali 'Guido Carli', in Rome. A total of 124 participants were split into the RW or LW groups according to their responses on the Explicit Political Orientation Questionnaire, and completed the experimental procedure. A set of 121 candidates from the total were included in the survey analysis (Exp1/Exp2 = 60/61) after exempting three who declared as being apolitical, while 118 candidates were included in the analysis of the economic behaviour (Exp1/Exp2 = 60/58) after exempting additional three who did not believe in the cover story. Participants were stratified for political orientation (LW/RW = 60/58), sex (M/F = 2/1) and age (LW/RW = 22.62/23.35 years).

## 2.3. Procedure

### 2.3.1. Participants' self-reported ideology

Participants' political orientation was measured via direct question, i.e. by asking them to self-place on a 7-point Likert-type item (1 = 'extremely left'; 2 = 'left'; 3 = 'centre- left'; 4 = 'centre-right'; 5 = 'right'; 6 = 'extremely right', 7 = 'apolitical'). According to the responses, participants were then split in two groups, with the LW-group consisting of voters who assigned scores from 1 to 3 on the self-placement scale, while the RW-group of voters who self-assigned scores ranging from 4 to 6. Those who rated themselves as apolitical (i.e. selected '7') were excluded from the experiment. We also introduced several control questions asking participants to specify the voted party preference at the last three consecutive elections (in order to make sure that selected participants had clear idea regarding their own political orientation), and to specify how often and to whom they talk about politics (in order to have an extra measure about participants' interest in politics). The questions regarding the past voting behaviour were introduced for the purpose of excluding participants who did not have clear idea about their political orientation and gave conflicting responses (e.g. self-identified as pro-left, while reported voting for pro-right political party, and vice versa).

### 2.3.2. Participants' moral values and social attitudes

Participants completed a computerized battery of questionnaires that helped to support the cover story and to provide information about their moral values and social attitudes. The questionnaire data helped to support the cover story and to provide information about the inferred ideology of participants. The following list of questionnaires was administered: (i) *Moral Foundations Questionnaire (MFQ)* [24] with 32 items organized in five core moral dimensions (Fairness/Reciprocity, Care/Harm Sensitivity,

In-group Loyalty, Authority and Purity/Sanctity) rated on a 6-point scale ranging from 0 (not at all relevant) to 5 (extremely relevant); (ii) *Economic System Justification Inventory (ESJI)* [25] designed to assess the tendency to 'legitimize economic inequality' through 17 items rated on a 1–9 Likert scale; (iii) *Right-Wing Authoritarianism (RWA)* [10] with 10 items measuring authoritarian submission, approval of authoritarian aggression and conventionalism on a 1–7 Likert scale; (iv) *Social Dominance Orientation (SDO)* [26] with 10 items measuring preference for in-group dominance on a 1–7 Likert scale; and (v) *Social Value Orientation (SVO)* [27] presented in six forced choices of self–other resource allocations and obtained with a slider-meter in order to yield a single score for the rank-order of participant's social preferences, i.e. the magnitude of concern that each participant had for others. Detailed account and statistics are provided in electronic supplementary material, 'Evaluation of Trustors'.

### 2.3.3. Cover story behind the trust game

A cover story was designed to deal with the fact that the real inclusion of a famous character in the role of a trustee in an experimental setting is hardly ever possible. It informed the participants about 'a mathematical model' that performed a psychological/ideological assessment of the real trustor (from their responses to the questionnaires), and on the basis of match/mismatch with the trustee's profile, calculated trustee's responses to specific offers. Constructed as such, the cover story allowed us to: ensure an *ecological reference* (by providing logical sense to a computer-guided economic exchange) and produce *credence* (by scaling-down partisan relations to one-on-one interaction, and zooming-in on follower–leader communication in the form of trustor–trustee series of actions and reactions).

### 2.3.4. Trustees

The experiments were conducted throughout 2014, in two consecutive periods (Exp 1: January–July and Exp 2: October–December), with the use of two famous characters in the role of trustee as a main difference. Specifically, the main experiment (Exp 1) used Silvio Berlusconi, former Prime Minister of Italy and leader of the centre-right parties' coalition, while the control experiment (Exp 2) used Piero Angela, a famous TV host. We conducted our main experiment over the course of one semester, for the purpose of avoiding unexpected political developments which could affect the public image and result in fluctuations in the public trust index towards Berlusconi. In order to control for a potential seasonal confound, we also consulted public electoral and political polls (managed by the Presidency of Ministers and the Italian Department of Information and Publishing[1]). Only one agency (IPR Marketing) provided reports throughout the whole year (December 2013, May 2014 and December 2014). According to these reports, trust index towards Berlusconi proved to be fairly consistent for the investigated period (25%, 23% and 20%, respectively). The obtained results confirmed that Berlusconi carried potential to arouse, sensitize and polarize the audience, and as such was suitable to be chosen as experimental stimulus. [4,28,29]. In order to choose a control stimulus who would be a better fit in terms of political neutrality and high popularity, we conducted a pilot online survey on an independent sample of 42 students, assessing two acclaimed national television hosts, Piero Angela (86 years) and Gerry Scotti (59 years). Both stimuli were selected for their high popularity and relatively close age to Berlusconi. Neither of the stimuli was an acclaimed politician or has been publicly associated with Italian politics. Participants were asked to decide if each of the TV hosts is taking a political side (i.e. to quantify the political orientation of the stimulus) and if so, which side they are taking (i.e. to qualify the political orientation of the stimulus). The former question was rated on a 5-point Likert scale (grading from 1 = 'not at all' to 5 = 'very much'), while the latter was rated on a 7-point Likert scale (1 = 'extremely left'; 2 = 'left'; 3 = 'centre-left'; 4 = 'centre-right'; 5 = 'right'; 6 = 'extremely right', 7 = 'apolitical'). In our analysis, we performed a unique index of the two measures by scaling participants' scores on each question, and calculating the overall political polarization of the stimulus as the distance from the centre (where 0.0 = 'centre' signifies the absence of political polarization, while 3.5 signifies highest political polarization). Two separate *t*-tests for each stimulus' ratings on political polarization were performed against zero, and both proved to be significant (Scotti: $t(39) = 7.14$, $p < 0.001$; Angela: $t(38) = 6.41$, $p < 0.001$), meaning that no stimulus could be perceived as absolutely neutral. However, the *t*-test for paired samples showed a significant difference on the ratings attributed to Scotti's and Angela's political polarization ($t(39) = 2.64$, $p < 0.05$), with Scotti receiving an average score of 0.51, while Piero Angela a score of 0.31. Considering that the complete absence of political

---

[1]Accessed through the website http://sondaggipoliticoelettorali.it/

polarization was rated as 0.0, while the maximal polarization was 3.5, both characters were minimally polarized. We opted for Angela as a significantly better-fitted control, both in terms of closer age and lower political polarization. In addition, we did not expect considerable fluctuation in the public perception of Angela, because he is a famous non-politician with long-term media presence, but stable public appearance. Hence, we were not restricted by the period and conducted our Exp. 2 in the second half of 2014. However, in order to control for the potential seasonal confound, we relied on our studies (since there are no available public perception polls for Angela). Namely, in our pilot (conducted at the beginning of 2014) and the experimental study (conducted towards the end of 2014), the participants on both sides of the political spectrum, did not differ significantly in their perception of Angela and considered him as a relatively neutral character. The final choice of control and main stimulus was further verified in each of the following dimensions: Dominance, Competence, Acquaintance, Positive and Negative Valence. As expected, both groups did not differ in the assessment of Angela, but they did in the assessment of Berlusconi on Competence, Acquaintance, Positive and Negative Valence. Importantly, there was no evident between-group distinction in the ratings for Dominance, since both groups assigned higher scores to Berlusconi on that dimension. Detailed account and statistics are provided in electronic supplementary material, 'Evaluation of Trustees'.

### 2.3.5. Trust game procedures

In our version of the TG, participants were asked to make an offer to the trustee (Silvio Berlusconi, the RW political leader; or Piero Angela, the famous TV host) in 15 rounds. In each round, participants could send to the trustee any amount between 1 and 10 euros. They knew that the invested amount would be quadrupled and that the trustee could decide to return half of the total or nothing at all. As a cover story, each participant was informed that the return offer of the trustee would correspond to their profile and actual behaviour, as determined by a simulation algorithm. In reality, the trustee return behaviour depended on whether the participants were unknowingly assigned to a pre-arranged trust (with trustee reciprocating in 80% of the trials) or distrust TG version (with trustee reciprocating in 20% of the trials). Details are provided in electronic supplementary material, 'Methods: Experimental design'. Even though participants performed 15 rounds, they were not informed about the total number of TG rounds in order to avoid the development of gain-benefit strategies. A detailed account is provided in electronic supplementary material, 'Methods: Experimental design'.

### 2.3.6. Manipulation check and briefing

Participants were told that one of the rounds will be randomly selected and paid in addition to the standard rate for participation in the experiment. In reality, each player received a fixed bonus rate due to the lack of a real mathematical model. The procedure concluded with a survey addressing participants' trust in the cover story without disclosing or suggesting the aim of the study in advance. The manipulation check procedure was carefully designed and it started with the following indirect questions: 'According to you, what is the scope of the experiment?' (to which participants provided written explanations) and 'Was the mathematical model predictive of Trustee's behaviour?' (to which they responded by using a 0–100 visual analogue scale (VAS)). The participants who indicated predictability of the mathematical model lower than 30% (the cut-off value), were again asked to provide written rationale for their reasoning. At the end, participants whose summary answers on all three questions implied that they guessed the real aim of the experiment, were directly asked whether they believe in the cover story. A negative response was taken as the main reason for exclusion of the participant from the analysis of the economic behaviour. The participants who were included (i.e. believed in the story about 'the mathematical model'), were surveyed regarding the reliability of 'their partner' in the game. Hence, they provided post-experimental evaluation of how un/trustworthy was the trustee during the game (on a 0–100 VAS). All participants were debriefed and informed about the background procedure and the real scope of the experiment.

## 3. Results

### 3.1. Self-reported ideology as indicator of economic behaviour

In order to investigate the level of initial trust expressed through the amount of investment in the first trial, we performed a $2 \times 2$ ANOVA with the Group (LW/RW) and Trustee (SB/PA) as between-

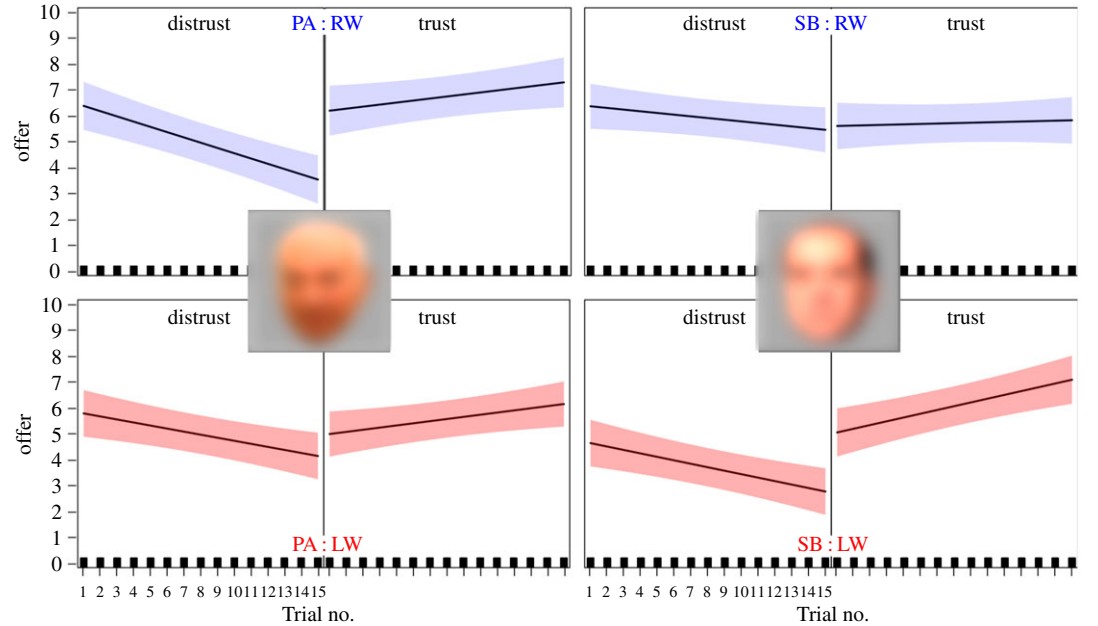

**Figure 2.** Illustration of participants' offers (in blue, those of the right-wingers; in red, those of the left-wingers) plotted according to the Trial no. (1–15), Group (LW/RW), TG Condition (trust/distrust) and Trustee (SB/PA). The coloured areas mark 95% CI.

subjects factors, while the initial investment was taken as a dependent variable. Contrary to our hypotheses regarding the initial investments, we did not find a significant double interaction ($F(1, 114) = 0.95$, $p = 0.331$), meaning that at the beginning of the game, the two groups did not differ significantly in the money they invested in either SB or PA.

To explore the overall dynamics of trust we employed linear mixed-effects modelling (LME) with the R package 'lme4', fit by maximum-likelihood $t$-tests using Satterthwaite approximations to degrees of freedom. This approach allowed us to thoroughly investigate the relationship between the economic behaviour indexed by the average amount of investment offered by participants at each trial and their self-reported political orientation (LW/RW). Therefore, we specified several models of increasing complexity with the simplest as a random-intercept model, allowing only between-subject variance in the average investment. The full model had the following fixed effects: Trial no. (1–15), Group (LW/RW), TG Condition (trust/distrust) and Trustee (SB/PA) (with interaction term). Participants were always treated as random effects with varying intercepts. The model selection relied on Akaike information criterion (AIC) and likelihood ratio (LR).

The results revealed that the full-interaction model was marked by both AIC drop from 8238.1 to 8235.8 ($\Delta$AIC = −2.3), and a significant change in the LR when compared with its nested model ($\chi^2(1) = 6.98$, $p < 0.01$). (For details see electronic supplementary material: 'Self-reported ideology as indicator of economic behaviour: full-interaction Model'.) Thus, the full-interaction model was selected as a best fit, showing that the economic behaviour of the trustor changed as a function of the interaction between trustee's and own political orientation, and their mutual economic exchange. The changes in trustor's behaviour unfolded throughout 15 exchanges, hinting at a considerable short-term plasticity of trust figure 2.

We also performed a separate LME analysis for each Group (LW/RW) and Trustee (SB/PA), with Trial no. (1–15) and TG Condition (trust/distrust) as fixed effects, and participants as random effects of varying intercepts, in order to observe the extent to which changes in trustors' economic behaviour could be attributed to the condition when playing with the two trustees (SB/PA). Results showed that for LW-participants the confidence intervals in the investigated interaction did not include zero when playing either with Berlusconi ($b = −0.27$, 95% CI; −0.36, −0.18) or with Angela ($b = −0.19$, 95% CI; −0.29, −0.10). This interaction indicated that there was a difference in the economic behaviour between the two TG Conditions in the LW-group regardless of whether they played with Berlusconi or Angela, meaning that they adjusted their behaviour (by increasing or decreasing their offers throughout the game) according to the unfolding of the putative behaviour of each trustee. By contrast, the RW-group did not differentiate in their behaviour during the task when playing with Berlusconi ($b = −0.08$, 95% CI; −0.19, 0.03) but they did when playing with Angela ($b = −0.27$, 95% CI; −0.38, −0.16) (figure 3).

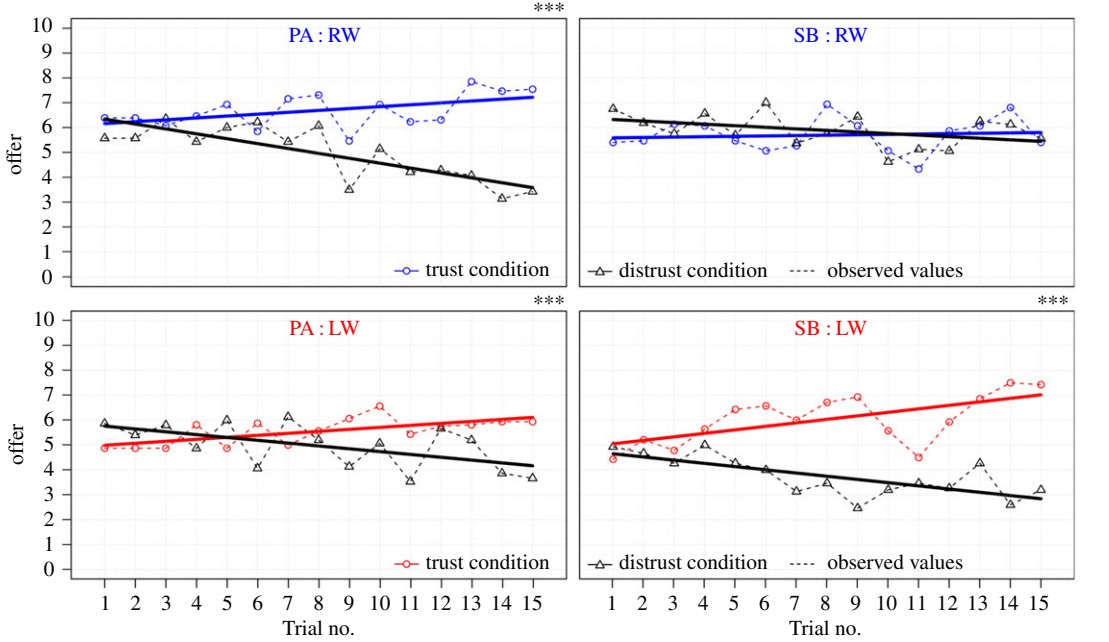

**Figure 3.** Chronological visualization of TG dynamics. The average amount of investment in euros (*y*-axis) is plotted against the sequence of trials (*x*-axis). The pairs of curves correspond to the trust/distrust conditions of the game, per each group (RW/LW) and trustee (SB/PA). The resulting trial-by-trial overview displays observed (dashed lines) and modelled/predicted economic behaviour (solid lines). ***$p < 0.001$.

In order to explore if participants' economic behaviour (throughout the game) was coherent with their overall impression about their partners' trustworthiness (measured at the end of the game), we ran a $2 \times 2 \times 2$ ANOVA with the Group (LW/RW), TG Condition (trust/distrust) and Trustee (SB/PA) as between-subjects factors (i.e. independent variables), and ratings on Perceived Trustworthiness (0–100 VAS) as a dependent variable. This analysis did not result in a significant triple interaction ($F(1, 109) = 0.88$, $p = 0.351$), showing that the two groups did not differ significantly in the explicit trustworthiness ratings of their respective partners and for the respective conditions. We did find a significant main effect of the trust/distrust TG condition ($F(1, 109) = 47.129$, $p < 0.001$), showing that both groups attributed higher/lower trustworthiness ratings to the respective version of the game. The post-experimental evaluation of the overall behaviour of the trustee confirms that participants were explicitly aware of the trustee's behaviour.

## 3.2. Group binding dimension as possible indicator of economic behaviour

In order to investigate the relationship between participants' economic behaviour and their potential group-leader related bias, we repeated the LME analysis by substituting self-reported Group (LW/RW) with data obtained from previously completed questionnaires on their moral values and social attitudes. First, we performed a principal component analysis (PCA) with varimax orthogonal rotation on nine variables with the following Cronbach reliability coefficients: Harm (0.81), Fairness (0.78), Loyalty (0.79), Authority (0.75), Purity (0.76) from the Moral Foundations Questionnaire (MFQ), and Right-Wing Authoritarianism (RWA = 0.72), Social Dominance Orientation (SDO = 0.73), Economic System Justification (ESJ = 0.74) and Social Value Orientation (SVO = 0.80). Overall, reliability analysis returned a general score of 0.79 with a range from 0.73 to 0.81 for separate items.

The analysis of the eigenvalues revealed a two-factor structure that explained 62.77% of the total variance. The first factor accounted for 41.23% of the variance and the variables with the highest loadings in this group are listed as follows: MFQ Purity (0.85), MFQ Authority (0.83), RWA (0.78) and MFQ Loyalty (0.70). The second factor was mainly loaded by: MFQ Fairness (0.85), SDO (−0.75), MFQ Harm (0.71), ESJ (−0.62). In line with previous literature on ideology-inferred differences from moral foundations and social attitudes [9,24,30], the first factor was named Group binding dimension, since it comprised binding moral values together with the RWA attitudes, thus tapping onto the social conformity construct. The second factor was named Social Equality Dimension, since it comprised the

remaining moral values concerned with fairness and harm reduction (individualizing MFQ variables), decreased proneness to hierarchy (SDO) and legitimization of economic inequality (ESJ), thus tapping onto the social equality construct. As previously demonstrated [12], the first factor is associated with group-centrism and possible group-related favouritism, so we decided to proceed with the investigation on the interaction between the degree of Group binding dimension and the economic behaviour of voters indexed by the average amount of investment.

Specifically, we decided to repeat the aforementioned LME with only one procedural modification: the self-reported political group in the full model, i.e. Group (RW/LW), was substituted with the score on the Group binding dimension (extracted with PCA from the questionnaires and used as a single-item measure i.e. independent variable), since the correlation between the two variables was statistically significant ($r = 0.71$, $p < 0.001$). Again, the full-interaction model was selected as a best fit, both in terms of overall AIC drop from 8316.4 to 8247.9 ($\Delta$AIC $= -68.5$) and marginal LR ($\chi^2(1) = 2.73$, $p = 0.098$). Thus, results showed that the economic behaviour of the trustor changed as a function of the interaction between trustor's score on Group binding dimension and the trustee's political orientation, unfolding over the course of the economic exchange. Mirroring previous results, the lack of modulation towards the conservative leader between two conditions of the TG was higher for people who scored higher on the Group binding dimension. (For details see the electronic supplementary material: 'Group binding dimension as indicator of economic behaviour: full-interaction model'.)

## 4. Discussion

The TG dynamics between the political leaders and their devoted followers occurs on a daily basis. The leaders usually assume a role of trustworthy subjects, by offering economic incentives (investments, credits and other financial benefits) to their trusting followers. In addition, the TG dynamics is also reflected in the fact that each voter is also a tax-payer (i.e. trustor), while the political leaders with governmental experience are decision-makers (i.e. trustees). They can choose to spend the budget for societal improvements (thus returning the trust and behaving trustworthy), or for their own benefits (thus behaving in an untrustworthy manner). Our approach allowed us to shed light on important aspects of these voter–leader interactions. We demonstrated that when it comes to supporting voters and their perception of an in-group political leader, bias may not manifest at the very beginning, but it can certainly become evident towards the end of a trust-based economic interaction. Specifically, we observed a clear difference in the overall behaviour between our two experimental groups when playing with an in-group/out-group political leader (SB), as compared to playing with a neutral control (PA). Namely, we found that while opposing voters (in our case LW-group) adjust the amount of their investment according to the behaviour of both SB and PA, supporting voters (in our case RW-group) failed to adjust when playing with SB, but did so when playing with PA. The lack of adjustment in the group who played with their own leader occurred despite the fact that they were indeed explicitly aware of his behaviour (as confirmed by the post-experimental evaluation of the overall behaviour of the trustee).

Our central finding (related to the lack of adjustment in the group who played with their un/trustworthy political leader) provides evidence beyond the existing literature on the intergroup ideological bias (i.e. the automatic preference for the members of one's own political group), guided by the preference for one's own political leader. Also, it supports the so-called epistemic spillover, a flawed heuristic of trusting like-minded political figures in non-political matters (in this case, economic trust decisions in an interactive game). Moreover, our findings suggest the mechanisms that are at the core of ideological intergroup biases, by highlighting the role of two important dimensions, especially prominent in RW-voters: *(i) the strong group-binding moral values*, and *(ii) the respect for the in-group authority figures (i.e. powerful leaders)*.

There are studies providing ample neural [31] and behavioural [32–34] evidence that voters from both sides of the political spectrum are not bound by cold rationality when making judgements about their political peers. Our results expand on previous literature showing that, at least on the explicit level, both groups of trustors were able to discriminate the behaviour of the trustee. Namely, they were able to discern whether the trustee reciprocated their offers, as indexed by the lack of significant differences between the groups in the assessment of both characters (PA and SB) and measured through the post-experimental evaluations of the trustors. However, we observed clear differences in the economic behaviour of voters who played with their in-group leader as indexed by the significant triple interaction (Group × Trustee × Condition) of the mixed-model analysis. Furthermore, a more in-depth analysis of moral values and social attitudes on our sample of voters (obtained with PCA on a

battery of self-reported surveys) suggests that the stronger admiration for the authority of one's leader, and the strong loyalty to one's group could be the leading mechanisms for the emergence of the bias [28,35]. These results are in line with the Social Identity Theory regarding expressive partisan identity [7], the Moral Foundation Theory (MFT) and the Right-Wing Authoritarianism Theory (RWA) [8–10,24]. According to MFT, the five basic moral foundations (harm/care, fairness/reciprocity, in-group/loyalty, respect/authority and purity/sanctity), collapse into two super-ordinate foundations. Specifically, the first couple of values are labelled as individualizing foundations (generally oriented towards protection and fair treatment of individuals), while the remaining three as binding foundations (focused on protection of the group, collectives, institutions). The binding foundations include: (i) patriotism and self-sacrifice for one's group (in-group loyalty); (ii) concerns about the importance of social order, traditions and respect for leadership (respect/authority); and (iii) the prevalence of spiritual over the carnal nature of humans (purity/sanctity). The political identity largely shapes people's moral foundations, with LW-voters endorsing the individualizing foundations, while RW-voters ascribing same or higher moral relevance to the binding foundations. The latter group of values have been examined in the context of immorality i.e. 'unacceptable behaviour, such as blind obedience and stigma' [9]. In our experiment, the observed lack of modulation in trust-based economic behaviour towards the conservative leader was stronger in people who scored higher on the well-described Group binding dimension. This dimension has been built upon the scores from the binding set of moral values (like loyalty for one's group and respect towards own authority figures with concern for religious purity matters), and RW-authoritarian social attitudes (i.e. authoritarian submission, tolerance for authoritarian aggression and conformism as defined in RW authoritarianism scale).

These findings also bring additional evidence for the existing claim that liberal–conservative differences in moral intuitions could be due to authoritarianism dimension, i.e. they happen because conservatives' greater valuation of their group, traditions, conventions and norms are attributable to their generally high levels of proneness to authority figures [12]. Indeed, in our experiment, the strong partisan affiliation emanated through the persona of a charismatic, dominant leader [36], even if incidental, was still the case when playing with SB. In relation to this, Schilke *et al.* have already demonstrated that the power-disadvantaged would be inclined to perceive the power-holders in a positive light, even when such belief is not supported by reliable evidence [37]. Such conclusions should hold special truth for individuals who show higher awareness of the hierarchical arrangements and attribute greater importance to authority figures, as is the case with RW-partisans. The observed behaviour of supporting voters when playing with their influential leader, is also in line with our earlier finding on the gaze-attracting power of SB. Namely, a RW-group was less able to suppress the automatic tendency to follow SB's gaze in comparison to a LW-group [4], an effect linked to his political power [28]. Improved understanding may come from studies using advanced tools that would allow more realistic interaction with a leader (like immersive virtual reality), and provide more insightful, computational models (via machine learning), or neurophysiologic correlates (via fMRI) for data analysis and interpretation.

In an analogous fashion, voters can 'use the perceived credibility of political figures as a heuristic to guide their evaluations' of what is right or wrong, and decisions of whether to trust or not, i.e. the effect known as 'epistemic spillover' [13,14]. Accordingly, our previous studies have already shown that the perceived similarity between voters' and leaders' personality, can influence even the basic cognitive processes of voters, such as attentional gaze capture [4,16,28]. Our current experiment provides more direct evidence, showing that voters' shared political ideology with an in-group political leader 'spills over' to their economic decision-making processes in a trust economic game (i.e. unrelated, non-political context).

The study casts light on a number of additional issues concerning the understanding of individuals and group dynamics. The first is the extent to which trust towards a political leader is defined by voter–leader interchange. The second is the extent to which the voter–leader interchange relies on the influence of *situational factors*, in our case an online and dynamic version of the TG. The third is the extent to which trust towards a political leader is determined by *dispositional factors*, in our case voters' moral values and social attitudes, leading to its sustenance and behavioural resistance to change. Our results show that depending on the Group binding dimension, the voter–leader relation embedded in trustor–trustee economic interaction can either be a dynamic construct with situational attributions, or a fairly static construct with dispositional attributions. Importantly, when it comes to voter–leader relations, trust can be enlightening for some, but it can be just as blinding for others. We might argue that it is intuitive to expect that one will act irrationally when s/he is ideologically biased. In fact, supporting voters (i.e. RW-group) behaved in line with this reasoning: they were not able to adjust to un/trustworthy behaviour of their in-group leader, i.e. Berlusconi. We might also argue that it is counterintuitive to expect that one will act rationally when s/he is ideologically biased. In our

experiment, opposing voters (i.e. LW-group) behaved rationally in spite of this reasoning: they adjusted their economic behaviour to the un/trustworthy behaviour of the out-group leader Berlusconi. The behaviour of LW-group might be attributed to the fact that they were not playing with an in-group leader. Here, we want to highlight the fact that the behaviour of the RW-group towards an in-group leader, does not call for parallels or conclusions regarding the behaviour of the LW-group towards a hypothetical in-group leader. In fact, at this point, it remains unclear whether an ideological asymmetry will emerge, even though groups differ in the Group binding dimension. Therefore, we strongly support replication of the experiment in a nation with a strong LW-leader. In Italy, we could not recognize a single LW-leader who embodies all required qualities to be considered as Berlusconi's equivalent. Namely, the dimensions which are deemed crucial to control in the choice of a popular LW-leader are the following: long-term political presence and governmental leadership experience (e.g. a leader of coalition party or a Prime Minister of the country); persisting popularity via continuous public media appearance; clear perception regarding leader's political orientation, i.e. a person with unambiguous political categorization. Indeed, all elected Italian LW-leaders differ from Berlusconi in something more than ideology. For instance, the Italian Prime Minister at the time of the experiment, the much younger LW-leader Matteo Renzi, was not elected by citizens, but chosen by the President of Republic with mandate to form a technical administration. Also, he lacked long-term political presence and media appearance. On the other hand, the career of the LW-politician Romano Prodi, is indeed marked with long-term political presence (Prime Minister in the periods of 1996–1998 and 2006–2008) which is comparable to Berlusconi's leadership experience (Prime Minister in the periods of 1994–1995, 2001–2006 and 2008–2011). However, he lacks ongoing public exposure and media appearance (since his decade-long retirement from Italian politics). Future studies with longitudinal explorations (spanning across multiple seasons) and diverse geopolitical settings (especially, countries with a strong left-wing equivalent) will shed important light on the subject. Specifically, studies on other leaders with matching years of political presence, leadership experience and media appearance can help to address the question about the power that an in-group (conservative or liberal) leader exerts on his followers.

In conclusion, by investigating direct voter–leader trust-based economic exchange, our study provides novel empirical evidence that the power of the political leader on his/her electors does not rely merely on the perceptions about his unprecedented authority, but also on biased perceptions of whether s/he is trustworthy. Ultimately, these findings can contribute to an advanced understanding of media campaigns and electoral outcomes. Thus, the study carries potential to help improve democracy in the modern-day human societies.

Ethics. The study involved only human participants, and no animals were employed for the purpose of this research. It was carried out in accordance with the 1964 Helsinki Declaration for Research Ethics. In addition, the study was approved by the Institutional Review Board at the Scientific Institute for Research Hospitalization and Health Care—Santa Lucia Foundation in Rome, Italy (IRB Protocol: CE/AG4-PROG.191-71). All participants have agreed to participate, and have signed an informed consent providing detailed account of the procedure.

Data accessibility. Our dataset and codes are publicly available at Open Science Framework repository and they can be retrieved from the following address: osf.io/nygs6/.

Authors' contributions. B.G., M.T.L., G.P., G.V.C. and S.M.A. designed research; B.G. performed research; B.G., M.T.L., G.P. and S.M.A planned analyses; B.G., M.T.L. and G.P. performed analyses; all authors wrote the paper and contributed to revisions.

Competing interests. The authors declare no competing interest with respect to the research, authorship, financing and publication of this article, or any other conflict of interest whatsoever.

Funding. The research was supported by the grants of national research interest (PRIN: Progetti di Ricerca di Rilevante Interesse Nazionale, Edit. 2015, Prot. 20159CZFJK and Edit. 2017, Prot. 2017N7WCLP), as well as the ERC Advanced Grant, 2017 (eHONESTY, 789058).

Acknowledgements. We are grateful to Maria Serena Panasiti for helpful suggestions, Adriano Acciarino and Gianluca Angeli for their help in data collection. Also, we want to express our gratitude to the anonymous reviewers for their diligent revision and substantial contribution in the improvement of the manuscript.

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
