## [Reviewer comments · Royal Society Open Science]

Review History

RSOS-180505.R0 (Original submission)

Review form: Reviewer 1

Is the manuscript scientifically sound in its present form?

Yes

Are the interpretations and conclusions justified by the results?

No

Is the language acceptable?

Yes

Is it clear how to access all supporting data?

Yes

Do you have any ethical concerns with this paper?

No

Have you any concerns about statistical analyses in this paper?

No

Recommendation?

Accept with minor revision (please list in comments)

Comments to the Author(s)

This manuscript compares the behavior of left-wing and right-wing voters in an economic trust game toward a well-known politician from the right (Berlusconi) and a well-known non-politician without a stated ideology (Angela). The authors find that left-wingers adjust their behavior toward both Angela and Berlusconi (the latter representing their political outgroup) as a function of the two figures' purported trustworthy vs. untrustworthy behavior. In contrast, right-wingers adjust their behavior toward Angela as a function of his trustworthy vs. untrustworthy behavior, but they do not adjust their behavior toward Berlusconi (their political ingroup) as a function of his trustworthiness. Right-wingers' consistent investment behavior toward Berlusconi is not reflected in their trustworthiness ratings of him, and it is suggested that their context-independent trust behavior toward Berlusconi can be attributed to heightened social conservatism and right-wing authoritarianism tendencies.

This is a well-written manuscript and it explores an interesting political question of when and how voters will trust and support politicians with valuable resources, which is relevant across many political contexts today. I also thought it was interesting to explore different facets of political preferences (MF, RWA, ESJ, SDO, SVO).

The use of the trust game for the purpose of this experiment is clear, but I have a few questions about the design that I believe could be clarified. First, I like the inclusion of a non-political figure as a control, but I was wondering why there was no left-wing politician included in the design. This seems important because the paper is framed as right-wingers "blindly" investing in their ingroup politicians whereas left-wingers do not, but the described study does not provide an opportunity to assess whether left-wingers would also "blindly" invest in their own ingroup politicians. I also think including a left-wing politician condition would increase the study's ecological validity. In the discussion section, it seems to be suggested that there was no equivalent left-wing politician to Berlusconi in Italy, so one was not included here. However, there are certainly known left-wing politicians in Italy, and excluding such a condition should at least be explained more. At minimum, if a left-wing politician condition cannot be included, I think the language about the ideological differences should be softened substantially.

Is it possible that Angela is perceived as a relatively left-wing figure, even if he is not a politician? On this point, it could be helpful to include ideological ratings of the two figures in addition to the other ratings in the SI "evaluation of trustees" table.

As a side point, I was also wondering if participants found the trustworthy and untrustworthy interactions with Berlusconi (and to a lesser extent, Angela) believable. I don't think this is critical to the manuscript, but is there any way to assess whether the trustworthy vs. untrustworthy conditions reflect subjects' preexisting perceptions of Berlusconi?

Could the measures used for the Explicit Political Orientation questionnaire be provided in the text? Considering this is the IV used in the primary analyses, I was hoping to understand how ideology was measured -- for example, was it a single self-placement item, a composite measure of different ideological dimensions, or a composite measure of positions on a variety of policies?

I think the model using the social conservatism factor is interesting and illuminates potential mechanisms by which the observed ideological differences may emerge. Would it be possible to include a similar model using the economic liberalism factor, perhaps in the SI? It seems a lost opportunity to measure these constructs, which likely also play a role in the economic behavior in this context, but not to explore them.

Finally, it would be helpful to include some discussion about the political implications of these results. It's not totally clear to me how one-to-one interactions and exchanges with political leaders are realistic, but perhaps the authors could consider other economic scenarios/games in which the investments are explicitly framed as donations and trustees' behavior is linked to some policy outcomes, for example.

Review form: Reviewer 2

Is the manuscript scientifically sound in its present form?

No

Are the interpretations and conclusions justified by the results?

No

Is the language acceptable?

Yes

Is it clear how to access all supporting data?

Yes

Do you have any ethical concerns with this paper?

No

Have you any concerns about statistical analyses in this paper?

Yes

Recommendation?

Reject

Comments to the Author(s)

I am not entirely sure what to make of this manuscript. It is interesting, but to me it seems more like a promising pilot study than a definitive study. I am not entirely sure what it means, psychologically (to research participants) to play a trust game with Berlusconi's "algorithm," but it does raise some intriguing questions. I am not convinced that such a strange situation really says much about "voter-leader interactions" (p. 7). My main concern, however, is that the sample is too small to support meaningful conclusions, and it is potentially problematic that there was no left-wing equivalent to Berlusconi included in the design of the experiment.

The final sample was $N = 118$ (p. 4), and (before exclusions) there were 58 rightists and 63 leftists altogether. There were two between-participants experimental conditions, which means that there were roughly 24-32 (or less, after exclusions) participants in each condition. I am afraid that these numbers are just too small to warrant the drawing of conclusions about rightists and leftists

in Italy. I have not performed any power analyses, but I suspect that the study is dramatically under-powered.

Furthermore, the failure to include a left-wing equivalent to Berlusconi (like Romano Prodi) means that we cannot really tell whether there was an ideological asymmetry in terms of trust behavior. It is possible that leftist participants would persist in trusting Prodi, but it is also possible that they would not. Until this question is answered, it is not clear to me what we can conclude from this study.

There are other concerns I have as well. If I am reading things correctly, the Berlusconi condition was run many months before the control condition (Angela). This means that there is a confound between time (or season) and interaction partner.

I also have some questions about the results (and how to interpret them). Why were there no differences between leftists and rightists in terms of (a) how much they invested in Berlusconi vs. Angela (p. 5), and (b) explicit trustworthiness ratings of Berlusconi vs. Angela (p. 6)? I find these null results to be surprising, and they make interpretation of the behavioral results more difficult, in my judgment.

In terms of theoretical issues, I don't think that this paradigm can tell us anything about whether liberals and conservatives differ in terms of cognitive rigidity (p. 7). I also don't know what it has to do with moral foundations theory, except to the extent that the behavior is tapping into authoritarianism. I am unsure about the connection to gaze-attraction as well.

Decision letter (RSOS-180505.R0)

31-May-2018

Dear Dr Gjoneska:

Manuscript ID RSOS-180505 entitled "In leaders we trust: right-wing electors blindly invest in their political leader in an economic trust game" which you submitted to Royal Society Open Science, has been reviewed. The comments from reviewers are included at the bottom of this letter.

In view of the criticisms of the reviewers, the manuscript has been rejected in its current form. However, a new manuscript may be submitted which takes into consideration these comments.

Please note that resubmitting your manuscript does not guarantee eventual acceptance, and that your resubmission will be subject to peer review before a decision is made.

Your resubmitted manuscript should be submitted by 28-Nov-2018. If you are unable to submit by this date please contact the Editorial Office.

Please note that Royal Society Open Science will introduce article processing charges for all new submissions received from 1 January 2018. Charges will also apply to papers transferred to Royal Society Open Science from other Royal Society Publishing journals, as well as papers submitted as part of our collaboration with the Royal Society of Chemistry (<http://rsos.royalsocietypublishing.org/chemistry>). If your manuscript is submitted and accepted for publication after 1 Jan 2018, you will be asked to pay the article processing charge, unless you request a waiver and this is approved by Royal Society Publishing. You can find out more about the charges at <http://rsos.royalsocietypublishing.org/page/charges>. Should you have any queries, please contact openscience@royalsociety.org.

Kind regards,
Andrew Dunn
Royal Society Open Science
openscience@royalsociety.org

on behalf of Dr Molly Crockett (Associate Editor) and Antonia Hamilton (Subject Editor)
openscience@royalsociety.org

Associate Editor Comments to Author (Dr Molly Crockett):

Dear Dr Gjoneska and colleagues,

I have now received comments from two expert reviewers. The comments from reviewers are included at the bottom of this letter.

In view of the criticisms of the reviewers, the manuscript has been rejected in its current form. However, a new manuscript may be submitted which takes into consideration these comments. In particular, we would be willing to consider a new manuscript that includes a larger sample size and a left-wing equivalent to Berlusconi as an additional control.

Reviewers' Comments to Author:

Reviewer: 1

Comments to the Author(s)

This manuscript compares the behavior of left-wing and right-wing voters in an economic trust game toward a well-known politician from the right (Berlusconi) and a well-known non-politician without a stated ideology (Angela). The authors find that left-wingers adjust their behavior toward both Angela and Berlusconi (the latter representing their political outgroup) as a function of the two figures' purported trustworthy vs. untrustworthy behavior. In contrast, right-wingers adjust their behavior toward Angela as a function of his trustworthiness vs. untrustworthy behavior, but they do not adjust their behavior toward Berlusconi (their political ingroup) as a function of his trustworthiness. Right-wingers' consistent investment behavior toward Berlusconi is not reflected in their trustworthiness ratings of him, and it is suggested that their context-independent trust behavior toward Berlusconi can be attributed to heightened social conservatism and right-wing authoritarianism tendencies.

This is a well-written manuscript and it explores an interesting political question of when and how voters will trust and support politicians with valuable resources, which is relevant across many political contexts today. I also thought it was interesting to explore different facets of political preferences (MF, RWA, ESJ, SDO, SVO).

The use of the trust game for the purpose of this experiment is clear, but I have a few questions about the design that I believe could be clarified. First, I like the inclusion of a non-political figure as a control, but I was wondering why there was no left-wing politician included in the design. This seems important because the paper is framed as right-wingers “blindly” investing in their ingroup politicians whereas left-wingers do not, but the described study does not provide an opportunity to assess whether left-wingers would also “blindly” invest in their own ingroup politicians. I also think including a left-wing politician condition would increase the study’s ecological validity. In the discussion section, it seems to be suggested that there was no equivalent left-wing politician to Berlusconi in Italy, so one was not included here. However, there are certainly known left-wing politicians in Italy, and excluding such a condition should at least be explained more. At minimum, if a left-wing politician condition cannot be included, I think the language about the ideological differences should be softened substantially.

Is it possible that Angela is perceived as a relatively left-wing figure, even if he is not a politician? On this point, it could be helpful to include ideological ratings of the two figures in addition to the other ratings in the SI “evaluation of trustees” table.

As a side point, I was also wondering if participants found the trustworthy and untrustworthy interactions with Berlusconi (and to a lesser extent, Angela) believable. I don’t think this is critical to the manuscript, but is there any way to assess whether the trustworthy vs. untrustworthy conditions reflect subjects’ preexisting perceptions of Berlusconi?

Could the measures used for the Explicit Political Orientation questionnaire be provided in the text? Considering this is the IV used in the primary analyses, I was hoping to understand how ideology was measured -- for example, was it a single self-placement item, a composite measure of different ideological dimensions, or a composite measure of positions on a variety of policies?

I think the model using the social conservatism factor is interesting and illuminates potential mechanisms by which the observed ideological differences may emerge. Would it be possible to include a similar model using the economic liberalism factor, perhaps in the SI? It seems a lost opportunity to measure these constructs, which likely also play a role in the economic behavior in this context, but not to explore them.

Finally, it would be helpful to include some discussion about the political implications of these results. It’s not totally clear to me how one-to-one interactions and exchanges with political leaders are realistic, but perhaps the authors could consider other economic scenarios/games in which the investments are explicitly framed as donations and trustees’ behavior is linked to some policy outcomes, for example.

Reviewer: 2

Comments to the Author(s)

I am not entirely sure what to make of this manuscript. It is interesting, but to me it seems more like a promising pilot study than a definitive study. I am not entirely sure what it means, psychologically (to research participants) to play a trust game with Berlusconi’s “algorithm,” but

it does raise some intriguing questions. I am not convinced that such a strange situation really says much about “voter-leader interactions” (p. 7). My main concern, however, is that the sample is too small to support meaningful conclusions, and it is potentially problematic that there was no left-wing equivalent to Berlusconi included in the design of the experiment.

The final sample was $N = 118$ (p. 4), and (before exclusions) there were 58 rightists and 63 leftists altogether. There were two between-participants experimental conditions, which means that there were roughly 24-32 (or less, after exclusions) participants in each condition. I am afraid that these numbers are just too small to warrant the drawing of conclusions about rightists and leftists in Italy. I have not performed any power analyses, but I suspect that the study is dramatically under-powered.

Furthermore, the failure to include a left-wing equivalent to Berlusconi (like Romano Prodi) means that we cannot really tell whether there was an ideological asymmetry in terms of trust behavior. It is possible that leftist participants would persist in trusting Prodi, but it is also possible that they would not. Until this question is answered, it is not clear to me what we can conclude from this study.

There are other concerns I have as well. If I am reading things correctly, the Berlusconi condition was run many months before the control condition (Angela). This means that there is a confound between time (or season) and interaction partner.

I also have some questions about the results (and how to interpret them). Why were there no differences between leftists and rightists in terms of (a) how much they invested in Berlusconi vs. Angela (p. 5), and (b) explicit trustworthiness ratings of Berlusconi vs. Angela (p. 6)? I find these null results to be surprising, and they make interpretation of the behavioral results more difficult, in my judgment.

In terms of theoretical issues, I don't think that this paradigm can tell us anything about whether liberals and conservatives differ in terms of cognitive rigidity (p. 7). I also don't know what it has to do with moral foundations theory, except to the extent that the behavior is tapping into authoritarianism. I am unsure about the connection to gaze-attraction as well.

Author's Response to Decision Letter for (RSOS-180505.R0)

See Appendix A.

RSOS-182023.R0

Review form: Reviewer 1

Is the manuscript scientifically sound in its present form?

No

Are the interpretations and conclusions justified by the results?

No

Is the language acceptable?

No

Is it clear how to access all supporting data?

Yes

Do you have any ethical concerns with this paper?

No

Have you any concerns about statistical analyses in this paper?

No

Recommendation?

Major revision is needed (please make suggestions in comments)

Comments to the Author(s)

I read the revised manuscript with interest, and I appreciate the authors' efforts to address previous comments on their original manuscript. Although I still think the study is interesting, I continue to have concerns about the design and the conclusions drawn from the results, and I think it would be ideal to address these.

First, I have given a great deal of thought to the authors' arguments regarding the exclusion of a left-wing politician, and I am still troubled by the lack of this condition. In the real world, it is virtually impossible to find politicians on the left and right who are equivalent on every conceivable dimension; that doesn't mean that there is no point in assessing reactions to existing politicians with appropriate consideration of the limitations. I appreciate the authors' stringent criteria for experimental control here, but the ecological validity they gain by including a real politician like Berlusconi is, to my mind, somewhat undermined by the exclusion of a politician across the aisle, such as the ones they identify, Prodi or Renzi. I don't think 100% recognition is necessary to be included, nor is complete political equivalence necessary. I think that the authors could make a much stronger case if they decided to include a condition with a left-wing politician and simply make adjustments (experimentally or statistically) based on the differences between Berlusconi and the left-wing politician. In addition, there are significant (and perhaps confounding) differences between Berlusconi and the non-politician control, Angela: the authors state that Berlusconi was perceived as quite polarizing, whereas Angela was perceived as popular and not very polarizing. This seems a more significant confounding difference to me than the visibility issue with respect to a potential left-wing politician condition.

Second, I don't think that my point about the implied asymmetry between right wing and left wing participants was a misunderstanding, as the authors state in their response. I understand the point they are trying to make that they do not mean to imply that left wingers do not blindly follow their leaders – just that right wingers seem to. However, there are many points in the manuscript that suggest an ideological asymmetry that I just don't think is supported by the data. For example, in the abstract, it is stated that "Results revealed that left-wing voters relied on the situation (trustee's behavior), while right-wing voters did not." This suggests an asymmetry between left wingers and right wingers, and it is not supported by the data – both left wingers and right wingers updated their investment behavior in response to the non-politician. (Indeed, I'm not even sure that it's fair to say that right-wingers are "blindly" following their leader; rather, it seems just as plausible that they are doing this strategically, since, as the authors point out, they are able to adjust their behavior in response to the non-politician.) Furthermore, much of the text suggests that the authors are exploring a balanced design (e.g., on p. 2: "we were able to capture the initial trust of LW vs. RW participants towards an in-group vs. out-group leader

(hence potential positive vs. negative bias toward him”), but the design simply tests an in-group politician for RW participants and an out-group politician for LW participants. At minimum, I think the language about the design and the parameters that are actually being tested would be better served with greater constraint and precision.

Finally, I had not noticed the seasonal confound of the conditions in this experiment in my initial review, but I think it would be helpful to see more explanation about why running the two conditions in totally different seasons is not a worrisome confound. If participants had been randomly assigned to condition within the same season of data collection, the authors would still not have had the concern that perceptions of Berlusconi would be changing. I think it would be helpful to demonstrate that there were no newsworthy events in terms of politics or entertainment during or between the two seasons of data collection that could have affected participants’ perceptions.

Review form: Reviewer 3

Is the manuscript scientifically sound in its present form?

No

Are the interpretations and conclusions justified by the results?

No

Is the language acceptable?

Yes

Is it clear how to access all supporting data?

No

Do you have any ethical concerns with this paper?

No

Have you any concerns about statistical analyses in this paper?

No

Recommendation?

Major revision is needed (please make suggestions in comments)

Comments to the Author(s)

I highly commend the author's efforts at responding, thoroughly, to the reviewers' comments and concerns, addressing many of their points and making necessary clarifications. However, I really hoped that these clarifications or some remedies to serious methodological problems were remedied within the manuscript itself. I have a few recommendations below that hopefully will push this work forward:

(1) I recommend replicating the findings while, within the same experiment, randomly assigning participants to experimental and control groups.

(2) Within the experimental groups, I highly recommend using a group that has a LW leader as well. As such, the design will be a 3x2 (3 target ideology: right-wing, left-wing, control; 2 participant ideology: left-wing or right wing). Or, alternatively, and as guided by much research

on ideology in social psychology, I highly recommend that you treat ideology as a continuous variable. The authors provide reasoning as to why a matched LW target is not accessible, but any proxy may be possible-- they don't need to be exactly matched.

(3) I recommend that the authors provide power calculations to determine their sample sizes moving forward. A highly powered study could accommodate for some minute shortcomings in the design (e.g., a fully matched LW target). It would be a great stretch to talk about ideological symmetry or asymmetry without having targets across the ideological spectrum.

(4) It would also be important to state specific hypotheses that are clear and state to the point: specify clearly what the IV, DV, and covariates are to the reader. I was consistently lost in terms of what the vast collection of variables (such as ESJ, RWA, etc.) were doing in there, considering that the authors used the single-item political ideology measure as the IV in their analyses. The predictions were unclear, which made the results hard to follow.

(5) It would be important for the authors to completely flesh out why adjustment/trustworthiness were considered to be proxies for the "cognitive rigidity" concept. Their theoretical links could benefit from guided clarity. I read too many names of concepts and mechanisms and did not see how those were manipulated or measured in the study, or how they were related to the relationship between ideology and trust. The mechanisms were technically not directly tested, but were rather hypothesized. Maybe further studies could elaborate on the mechanisms.

I anticipate this work will be pushed in the right direction with a more refined design, more data, and clear mechanisms. This is an excellent starting point, and I wish the authors the best of luck moving forward!

Decision letter (RSOS-182023.R0)

20-Feb-2019

Dear Dr Gjoneska,

The Subject Editor assigned to your paper ("In leaders we trust: right-wing electors blindly invest in their political leader in an economic trust game") has now received comments from reviewers. We would like you to revise your paper in accordance with the referee and Associate Editor suggestions which can be found below (not including confidential reports to the Editor). Please note this decision does not guarantee eventual acceptance.

Please submit a copy of your revised paper before 15-Mar-2019. Please note that the revision deadline will expire at 00.00am on this date. If we do not hear from you within this time then it will be assumed that the paper has been withdrawn. In exceptional circumstances, extensions may be possible if agreed with the Editorial Office in advance. We do not allow multiple rounds of revision so we urge you to make every effort to fully address all of the comments at this stage. If deemed necessary by the Editors, your manuscript will be sent back to one or more of the original reviewers for assessment. If the original reviewers are not available we may invite new reviewers.

To revise your manuscript, log into <http://mc.manuscriptcentral.com/rsos> and enter your Author Centre, where you will find your manuscript title listed under "Manuscripts with

Decisions." Under "Actions," click on "Create a Revision." Your manuscript number has been appended to denote a revision. Revise your manuscript and upload a new version through your Author Centre.

When submitting your revised manuscript, you must respond to the comments made by the referees and upload a file "Response to Referees" in "Section 6 - File Upload". Please use this to document how you have responded to each of the comments, and the adjustments you have made. In order to expedite the processing of the revised manuscript, please be as specific as possible in your response.

- Ethics statement

- Data accessibility

If you wish to submit your supporting data or code to Dryad (<http://datadryad.org/>), or modify your current submission to dryad, please use the following link:
<http://datadryad.org/submit?journalID=RSOS&manu=RSOS-182023>

- Competing interests

- Authors' contributions

AB carried out the molecular lab work, participated in data analysis, carried out sequence alignments, participated in the design of the study and drafted the manuscript; CD carried out the statistical analyses; EF collected field data; GH conceived of the study, designed the study,

coordinated the study and helped draft the manuscript. All authors gave final approval for publication.

- Acknowledgements

- Funding statement

Kind regards,

Andrew Dunn

on behalf of Dr Antonia Hamilton (Subject Editor)

Associate Editor Comments to Author:

Thank you for resubmitting. The paper has been assessed by two reviewers, though, unfortunately, both identify a substantial number of matters that must be addressed before the paper could be considered acceptable for publication. With this in mind, we'd like you to revise the manuscript to address the reviewers' concerns. Please note that you will not be granted a second round of revisions, so do ensure you do all you can to resolve the reviewers' feedback: if they remain unsatisfied after revision, we will, with regret, be forced to reject the paper from further consideration.

Reviewer comments to Author:

Reviewer: 1

Comments to the Author(s)

I read the revised manuscript with interest, and I appreciate the authors' efforts to address previous comments on their original manuscript. Although I still think the study is interesting, I continue to have concerns about the design and the conclusions drawn from the results, and I think it would be ideal to address these.

First, I have given a great deal of thought to the authors' arguments regarding the exclusion of a left-wing politician, and I am still troubled by the lack of this condition. In the real world, it is virtually impossible to find politicians on the left and right who are equivalent on every conceivable dimension; that doesn't mean that there is no point in assessing reactions to existing politicians with appropriate consideration of the limitations. I appreciate the authors' stringent criteria for experimental control here, but the ecological validity they gain by including a real politician like Berlusconi is, to my mind, somewhat undermined by the exclusion of a politician across the aisle, such as the ones they identify, Prodi or Renzi. I don't think 100% recognition is necessary to be included, nor is complete political equivalence necessary. I think that the authors could make a much stronger case if they decided to include a condition with a left-wing politician and simply make adjustments (experimentally or statistically) based on the differences between

Berlusconi and the left-wing politician. In addition, there are significant (and perhaps confounding) differences between Berlusconi and the non-politician control, Angela: the authors state that Berlusconi was perceived as quite polarizing, whereas Angela was perceived as popular and not very polarizing. This seems a more significant confounding difference to me than the visibility issue with respect to a potential left-wing politician condition.

Second, I don't think that my point about the implied asymmetry between right wing and left wing participants was a misunderstanding, as the authors state in their response. I understand the point they are trying to make that they do not mean to imply that left wingers do not blindly follow their leaders – just that right wingers seem to. However, there are many points in the manuscript that suggest an ideological asymmetry that I just don't think is supported by the data. For example, in the abstract, it is stated that “Results revealed that left-wing voters relied on the situation (trustee's behavior), while right-wing voters did not.” This suggests an asymmetry between left wingers and right wingers, and it is not supported by the data – both left wingers and right wingers updated their investment behavior in response to the non-politician. (Indeed, I'm not even sure that it's fair to say that right-wingers are “blindly” following their leader; rather, it seems just as plausible that they are doing this strategically, since, as the authors point out, they are able to adjust their behavior in response to the non-politician.) Furthermore, much of the text suggests that the authors are exploring a balanced design (e.g., on p. 2: “we were able to capture the initial trust of LW vs. RW participants towards an in-group vs. out-group leader (hence potential positive vs. negative bias toward him)”), but the design simply tests an in-group politician for RW participants and an out-group politician for LW participants. At minimum, I think the language about the design and the parameters that are actually being tested would be better served with greater constraint and precision.

Finally, I had not noticed the seasonal confound of the conditions in this experiment in my initial review, but I think it would be helpful to see more explanation about why running the two conditions in totally different seasons is not a worrisome confound. If participants had been randomly assigned to condition within the same season of data collection, the authors would still not have had the concern that perceptions of Berlusconi would be changing. I think it would be helpful to demonstrate that there were no newsworthy events in terms of politics or entertainment during or between the two seasons of data collection that could have affected participants' perceptions.

Reviewer: 3

Comments to the Author(s)

I highly commend the author's efforts at responding, thoroughly, to the reviewers' comments and concerns, addressing many of their points and making necessary clarifications. However, I really hoped that these clarifications or some remedies to serious methodological problems were remedied within the manuscript itself. I have a few recommendations below that hopefully will push this work forward:

- (1) I recommend replicating the findings while, within the same experiment, randomly assigning participants to experimental and control groups.
- (2) Within the experimental groups, I highly recommend using a group that has a LW leader as well. As such, the design will be a 3x2 (3 target ideology: right-wing, left-wing, control; 2 participant ideology: left-wing or right wing). Or, alternatively, and as guided by much research on ideology in social psychology, I highly recommend that you treat ideology as a continuous variable. The authors provide reasoning as to why a matched LW target is not accessible, but any proxy may be possible-- they don't need to be exactly matched.

(3) I recommend that the authors provide power calculations to determine their sample sizes moving forward. A highly powered study could accommodate for some minute shortcomings in the design (e.g., a fully matched LW target). It would be a great stretch to talk about ideological symmetry or asymmetry without having targets across the ideological spectrum.

(4) It would also be important to state specific hypotheses that are clear and state to the point: specify clearly what the IV, DV, and covariates are to the reader. I was consistently lost in terms of what the vast collection of variables (such as ESJ, RWA, etc.) were doing in there, considering that the authors used the single-item political ideology measure as the IV in their analyses. The predictions were unclear, which made the results hard to follow.

(5) It would be important for the authors to completely flesh out why adjustment/trustworthiness were considered to be proxies for the "cognitive rigidity" concept. Their theoretical links could benefit from guided clarity. I read too many names of concepts and mechanisms and did not see how those were manipulated or measured in the study, or how they were related to the relationship between ideology and trust. The mechanisms were technically not directly tested, but were rather hypothesized. Maybe further studies could elaborate on the mechanisms.

I anticipate this work will be pushed in the right direction with a more refined design, more data, and clear mechanisms. This is an excellent starting point, and I wish the authors the best of luck moving forward!

Author's Response to Decision Letter for (RSOS-182023.R0)

See Appendix B.

RSOS-182023.R1 (Revision)

Review form: Reviewer 1

Is the manuscript scientifically sound in its present form?

Yes

Are the interpretations and conclusions justified by the results?

No

Is the language acceptable?

Yes

Is it clear how to access all supporting data?

Yes

Do you have any ethical concerns with this paper?

No

Have you any concerns about statistical analyses in this paper?

No

Recommendation?

Accept with minor revision (please list in comments)

Comments to the Author(s)

I really appreciate the authors' thoughtful consideration of the previous comments and their efforts to revise the manuscript accordingly. I believe that the paper is improved, and I hope the authors share this view. I do think there are a few areas in the paper that could use a little more clarity, and I hope the authors will consider these points.

(1) The added reasoning about the season/condition confound in the experiment is helpful, but it does not fully reassure the reader that there is not a seasonal (perception) confound with the treatment conditions that could be potentially affecting participants' behavior. I think the clearest way to show that the season of experimental administration did not affect the experiment would be to show (relative) invariability in public perceptions of both Berlusconi and Angela over the full span of the experiment, January to December (not just Berlusconi and not just January to June). If the authors have access to such perception polls, it would do a lot to address the confound.

(2) I think the reported group binding model is not as informative as it could be. It seems that the purpose of including this analysis is to demonstrate that this group binding dimension helps to explain why RW participants might be exhibiting this trusting behavior of Berlusconi. But the included model simply shows that people who are higher in group binding psychological preferences also exhibit this trusting behavior. I think it would make more sense to include these psychological variables with (rather than instead of) the ideology variable in the model to see if the psychological variables help to explain the relationship between ideology and behavior. Furthermore, it would be even more informative – given these variables were measured – to include not only the group binding measure but also the social equality measure in the model to test whether the group binding measure does a better job of explaining the variance than preferences for equality, as the authors may be suggesting. At minimum, it seems plausible that participants' economic preferences and their perceptions of Berlusconi's economic positions could play a role in their experimental behavior.

(3) A very minor point: How correlated was participants' voting behavior in the past three elections with their explicit political orientation (p. 15)? It's stated that this question was included to verify participants' understanding of their own ideology, but how this information was used would be even more helpful.

(4) Finally, as I mentioned in my initial review, it would be helpful to include some discussion about the political implications of these results. That is, how does this simulated one-on-one trust game exchange with a famous politician (which is unlikely to happen in the real world) reflect realistic voter behavior and experience? It would be really helpful to make the potential links between the experimental context and actual political outcomes more explicit.

Decision letter (RSOS-182023.R1)

08-Aug-2019

Dear Dr GJoneska,

On behalf of the Editors, I am pleased to inform you that your Manuscript RSOS-182023.R1 entitled "Bound to the group and blinded by the leader: ideological leader-follower dynamics in a trust economic game" has been accepted for publication in Royal Society Open Science subject to minor revision in accordance with the referee suggestions. Please find the referees' comments at the end of this email.

The reviewers and Subject Editor have recommended publication, but also suggest some minor revisions to your manuscript. Therefore, I invite you to respond to the comments and revise your manuscript.

- Ethics statement

- Data accessibility

If you wish to submit your supporting data or code to Dryad (<http://datadryad.org/>), or modify your current submission to dryad, please use the following link:
<http://datadryad.org/submit?journalID=RSOS&manu=RSOS-182023.R1>

- Competing interests

- Authors' contributions

- Acknowledgements

- Funding statement

Because the schedule for publication is very tight, it is a condition of publication that you submit the revised version of your manuscript before 17-Aug-2019. Please note that the revision deadline will expire at 00.00am on this date. If you do not think you will be able to meet this date please let me know immediately.

Kind regards,

on behalf of Professor Antonia Hamilton (Subject Editor)
openscience@royalsociety.org

Reviewer comments to Author:

Reviewer: 1
Comments to the Author(s)

I really appreciate the authors' thoughtful consideration of the previous comments and their efforts to revise the manuscript accordingly. I believe that the paper is improved, and I hope the authors share this view. I do think there are a few areas in the paper that could use a little more clarity, and I hope the authors will consider these points.

(1) The added reasoning about the season/condition confound in the experiment is helpful, but it does not fully reassure the reader that there is not a seasonal (perception) confound with the treatment conditions that could be potentially affecting participants' behavior. I think the clearest way to show that the season of experimental administration did not affect the experiment would be to show (relative) invariability in public perceptions of both Berlusconi and Angela over the full span of the experiment, January to December (not just Berlusconi and not just January to June). If the authors have access to such perception polls, it would do a lot to address the confound.

(2) I think the reported group binding model is not as informative as it could be. It seems that the purpose of including this analysis is to demonstrate that this group binding dimension helps to explain why RW participants might be exhibiting this trusting behavior of Berlusconi. But the included model simply shows that people who are higher in group binding psychological preferences also exhibit this trusting behavior. I think it would make more sense to include these psychological variables with (rather than instead of) the ideology variable in the model to see if the psychological variables help to explain the relationship between ideology and behavior. Furthermore, it would be even more informative – given these variables were measured – to include not only the group binding measure but also the social equality measure in the model to test whether the group binding measure does a better job of explaining the variance than preferences for equality, as the authors may be suggesting. At minimum, it seems plausible that

participants' economic preferences and their perceptions of Berlusconi's economic positions could play a role in their experimental behavior.

(3) A very minor point: How correlated was participants' voting behavior in the past three elections with their explicit political orientation (p. 15)? It's stated that this question was included to verify participants' understanding of their own ideology, but how this information was used would be even more helpful.

(4) Finally, as I mentioned in my initial review, it would be helpful to include some discussion about the political implications of these results. That is, how does this simulated one-on-one trust game exchange with a famous politician (which is unlikely to happen in the real world) reflect realistic voter behavior and experience? It would be really helpful to make the potential links between the experimental context and actual political outcomes more explicit.

Author's Response to Decision Letter for (RSOS-182023.R1)

See Appendix C.

Decision letter (RSOS-182023.R2)

22-Aug-2019

Dear Dr Gjoneska,

I am pleased to inform you that your manuscript entitled "Bound to the group and blinded by the leader: Ideological leader-follower dynamics in a trust economic game" is now accepted for publication in Royal Society Open Science.

on behalf of Prof Antonia Hamilton (Subject Editor)
openscience@royalsociety.org

Follow Royal Society Publishing on Twitter: [@RSocPublishing](https://twitter.com/RSocPublishing)
Follow Royal Society Publishing on Facebook:
<https://www.facebook.com/RoyalSocietyPublishing.FanPage/>
Read Royal Society Publishing's blog: <https://blogs.royalsociety.org/publishing/>

Appendix A

SAPIENZA
UNIVERSITÀ DI ROMA

RESPONSE TO REVIEWERS

REVIEWER No.1

“This manuscript compares the behavior of left-wing and right-wing voters in an economic trust game toward a well-known politician from the right (Berlusconi) and a well-known non-politician without a stated ideology (Angela). The authors find that left-wingers adjust their behavior toward both Angela and Berlusconi (the latter representing their political outgroup) as a function of the two figures’ purported trustworthy vs. untrustworthy behavior. In contrast, right-wingers adjust their behavior toward Angela as a function of his trustworthy vs. untrustworthy behavior, but they do not adjust their behavior toward Berlusconi (their political ingroup) as a function of his trustworthiness. Right-wingers’ consistent investment behavior toward Berlusconi is not reflected in their trustworthiness ratings of him, and it is suggested that their context-independent trust behavior toward Berlusconi can be attributed to heightened social conservatism and right-wing authoritarianism tendencies.

This is a well-written manuscript and it explores an interesting political question of when and how voters will trust and support politicians with valuable resources, which is relevant across many political contexts today. I also thought it was interesting to explore different facets of political preferences (MF, RWA, ESJ, SDO, SVO).”

We would like to thank this reviewer for the very precise overview of our study. We are very happy to find that our story is successfully conveyed, and that the study is positively evaluated by the reviewer.

Comment No.1

“First, I like the inclusion of a non-political figure as a control, but I was wondering why there was no left-wing politician included in the design. This seems important because the paper is framed as right-wingers “blindly” investing in their ingroup politicians whereas left-wingers do not, but the described study does not provide an opportunity to assess whether left-wingers would also “blindly” invest in their own ingroup politicians. I also think including a left-wing politician condition would increase the study’s ecological validity. In the discussion section, it seems to be suggested that there was no equivalent left-wing politician to Berlusconi in Italy, so one was not included here. However, there are certainly known left-wing politicians in Italy, and excluding such a condition should at least be explained more. At minimum, if a left-wing politician condition cannot be included, I think the language about the ideological differences should be softened substantially.”

First, we want to thank the reviewer for pointing at a crucial aspect related to the best control stimulus for Berlusconi. We will delve into the reasons for our decision to include a famous non-politician instead of a famous left-wing politician, as soon as we clarify possible misunderstandings related to the following comment: “the paper is framed as right-wingers ‘blindly’ investing in their ingroup politicians whereas left-wingers do not, but the described study does not provide an opportunity to assess whether left-wingers would also “blindly” invest in their own ingroup politicians”. We underline the part of the quoted segment which was not included, nor implied in our paper. Namely, a conclusion about a certain behavior regarding RW-followers toward their in-group leader, is not mutually exclusive with (or in any way indicative of) the behavior of LW-followers toward their in-group leader. Therefore, the result that RW-followers “blindly” invest in an in-group leader, does not automatically imply that LW-followers behave in opposite manner, when confronted with their in-group leader in an economic trust game. As the reviewer rightly observes, we did not include a LW-leader as a control stimulus, so we were careful not to make any such claims upon shaping of the paper. This is evident in opening/closing segments of the paper i.e., the title and the discussion as follows:

- The original title “Right-wing electors blindly invest in their leader in the economic trust game” highlights the most prominent result related solely to the behavior of the RW-group i.e., the fact that they displayed a constant pattern of behavior towards their leader regardless of the trust/distrust version of the game. However, the title does not suggest that such behavior is juxtaposed (or in any way comparable) to the behavior of LW-group, had they been placed in a similar situation.

- When discussing the results, we reason that it might be “intuitive to expect that one will act irrationally when s/he is ideologically biased. In fact, RW-voters behaved in line with this reasoning: they did not adjust to trustworthy/untrustworthy behavior of their in-group leader, i.e., Berlusconi”. However, we also argue that it might be “counter-intuitive to expect that one will act rationally when s/he is ideologically biased. In our experiment, LW-voters behaved rationally in spite of this reasoning: they adjusted their economic behavior to the trustworthy/untrustworthy behavior of the out-group leader Berlusconi”. Here again, we are careful to interpret our results only in the light of the collected evidence. We did not extend the comments to a hypothetical situation of LW-group playing with an in-group leader. In fact, we go on to speculate that such behavior might be attributed to “the fact that they were not playing with an in-group leader” which, in a way, is opposite to the claim that reviewer makes in his quoted segment.

We added a paragraph in the discussion of the revised manuscript to highlight the point (page 9, lines 17-21).

Moreover, to further confirm our decision of not including a left-wing politician, in the concluding remarks of the paper we clearly affirm that “in Italy at present, we could not recognize a single LW-leader with same level of popularity as Berlusconi”. We agree with reviewer’s suggestion that “including a left-wing politician condition would increase the study’s ecological validity”, but we don’t entirely agree with his/her supporting argument that “there are certainly known left-wing politicians in Italy”. According to us, the dimensions that are deemed crucial to control in the choice of a renowned/popular LW-leader (i.e., Berlusconi’s equivalent) are the following:

- Long-term political presence and governmental leadership experience (for example, a leader of a coalition/party or a Prime Minister of the country);*
- Persisting popularity via continuous public exposure and media appearance;*

- *Clear perception of leader's political orientation i.e., a person with unambiguous political categorization.*

In Italy we could not find a LW-politician who embodies all required qualities. Indeed, all of the elected candidates did differ from Berlusconi in something more than ideology. For instance, the Italian Prime Minister at the time of the experiment, the much younger LW-leader Matteo Renzi, was not elected by citizens but chosen by the President of Republic instead, with mandate to form a technical administration. Also he lacked long-term political presence and media appearance. On the other hand, the career of the LW-politician Romano Prodi, is indeed marked with long-term political presence (Prime Minister in the periods of 1996-1998 and 2006-2008) which is comparable to Berlusconi's leadership experience (Prime Minister in the periods of 1994-1995, 2001-2006 and 2008-2011), but he lacks ongoing public exposure and media appearance (since his decade-long retirement from Italian politics).

In addition, the politicians who are currently popular are not as clearly recognized or politically categorized, as Berlusconi. This claim is supported by data from a recent experiment we conducted in 2018, for a separate study. Namely, in an independent sample of 120 RW/LW Italian voters, we tested their level of recognition and political categorization of the most prominent Italian politicians belonging to the LW-coalition, RW-coalition, or the party called "Movimento Cinque Stelle". Specifically, participants were asked to observe the faces of 30 liberals, 28 conservatives, and 9 popular politicians from the "Movimento Cinque Stelle". After the exposure, they were asked to attribute a name to each face (recognition) and rate the politicians' ideology (political categorization). This bears similarity to the automatic evaluation by our students participating in the trust economic game, since we also included a picture of the trustee (which requires prior recognition). Statistical analysis of these data confirm that the only politician who was unanimously recognized and equally categorized as conservative leader by 100% of the respondents, was indeed Berlusconi. On the other hand, the LW-politicians who were either unanimously recognized (like Matteo Renzi), or politically categorized (like Nichi Vendola) by all respondents, did not receive full recognition/categorization respectively. Also, we would like to highlight the fact that these results are

collected in 2018, when Renzi was already recognized as a former Prime Minister (period of 2014-2016), while Vendola became prominent LW-politician (former president of Regione Puglia, 2005-2010, and leader of the national left-wing party "Sinistra Ecologia e Libertà, 2010-2016). Since we couldn't rely entirely on the prominent LW-politicians we also analyzed the political recognition/categorization for the representatives from "Movimento Cinque Stelle". As expected, their most prominent candidates had very high recognition, but very ambiguous political categorization.

In summary, we believe that the political settings in Italy at the time of data collection did not allow us to find any LW-match to Berlusconi. However, we strongly support the idea of replicating the study in a different geopolitical context and recommend "that future studies from countries where a left-wing equivalent (with matching years of political presence, leadership experience, and media appearance) exists, can help to address the question about the power of the leader on both LW- and RW-voters".

Upon reviewer's suggestion, in the discussion of the manuscript we included a summary of this response, containing the explanations about the most prominent Italian LW-politicians, and our decision to sustain from employing them as stimuli in our experiment (page 9, lines 22-40).

Comment No. 2

“Is it possible that Angela is perceived as a relatively left-wing figure, even if he is not a politician? On this point, it could be helpful to include ideological ratings of the two figures in addition to the other ratings in the SI ‘Evaluation of trustees’ table.”

We thank the reviewer for posing such insightful question which provides us with opportunity to offer more in-depth account of our study.

*In the preparatory phase of the experiment, we conducted an online survey on an independent sample of 42 students, assessing two acclaimed national television hosts, Piero Angela (86yr) and Gerry Scotti (59yr). Both stimuli were selected for their high popularity and relatively close age to Berlusconi. Neither of the stimuli was an acclaimed politician or has been publicly associated with the Italian politics. In order to choose a control stimulus who would be a better fit in terms of political neutrality participants were asked to decide if each of the TV-hosts is taking a political side (i.e., to quantify the political orientation of the stimulus) and if so, which side they are taking (i.e., to qualify the political orientation of the stimulus). The former question was rated on a 5-point Likert-scale (grading from 1= “not at all” to 5 = “very much”), while the later was rated on a 7-point Likert-scale (1=“extremely left”; 2=“left”; 3=“center-left”; 4=“center-right”; 5=“right”; 6=“extremely right”, 7=“apolitical”). In our analysis, we performed a unique index of the two measures by scaling participants’ scores on each question, and calculating the overall political polarization of the stimulus as the distance from the center (where 0.0 = “center” signifies the absence of political polarization, while 3.5 signifies highest political polarization). Two separate *t*-tests for each stimulus’ ratings on political polarization were performed against zero, and both proved to be significant (Scotti: $t(39) = 7.14, p < .001$; Angela: $t(38) = 6.41, p < .001$), meaning that no stimulus could be perceived as absolutely neutral. However, the *t*-test for paired samples showed a significant difference on the ratings attributed to Scotti’s and Angela’s political polarization ($t(39) = 2.64, p < .05$), with Scotti receiving an average score of 0.51, while Piero Angela a score of 0.31. Considering that the complete absence of political polarization was rated as 0.0, while the*

maximal polarization was 3.5, both characters were minimally polarized. We opted for Angela as a significantly better-fitted control, both in terms of closer age and lower political polarization.

In order to directly address the reviewer’s comment, we also analyzed the ideological scores attributed to the chosen control stimulus (Piero Angela), by the sample participating in the experiment (RW- and LW-group of students). Here, again we employed a 7-point Likert scale in order to qualify Angela’s political orientation (1=“extremely left”; 2=“left”; 3=“center-left”; 4=“center-right”; 5=“right”; 6=“extremely right”, 7=“apolitical”). The responses marked as “7” were substituted with “0” (signifying no political orientation whatsoever), while the rest were scaled in the range from -2.5 (“extremely left”) to 2.5 (“extremely right”). The results show that the sample of participating students, again rated Angela similarly, with an average score of -0.31, suggesting that indeed he was considered as a figure with low political polarization. In addition, there were no statistical differences between the groups (RW/LW) on the political categorization of Angela ($F(119) = 3.32$; $p = .071$). Hence, both had similar perceptions of Angela and neither LW-voters nor RW-voters perceived him as more politically inclined toward the liberal/conservative ideologies. As insightfully suggested by the reviewer, we included the results regarding the assessment of the control stimulus in a separate table (added in the SI “Evaluation of Trustees”). As evident from the Tab.1, the obtained result may be indicative of the notion that no group perceived him as a potential in-group/out-group member/leader.

Political Orientation: Evaluation of Piero Angela						
Group	F(1, 119) = 3.32	N°	Mean	SD	+95% CI	-95% CI
	p-value					
LW	.071	63	-0.44	0.80	-0.64	-0.23
RW		58	-0.17	0.79	-0.38	0.04

Tab. 1 One-way ANOVA on 121 participants with the Group (RW/LW) as between-subjects factor, and the ratings regarding Piero’s Angela Political Orientation as dependent variable.

This result is also mirrored in the economic behavior of participants at the beginning of the Trust game and throughout its entire course. Namely, initial investments in both groups did not differ significantly for the amounts of money they sent towards Angela or Berlusconi. Indeed, the 2 x 2 ANOVA with the Group (LW/RW) and Trustee (SB/PA) as between-subjects factors, and the initial investment (1-10 euro) as a dependent variable, shows no statistical difference between the groups for each of the Trustee ($F(1, 114) = .95, p = .331$). Hence, the result can be interpreted as a lack of preexisting in-group/out-group bias toward either Angela or Berlusconi. Furthermore, both groups adjusted their economic behavior throughout the game to the trust/distrust condition when they played with Angela, as evident in Fig. 3 of the manuscript showing the linear mixed-effects analysis for each Group (LW/RW) playing with each Trustee (SB/PA), with Trial No (1-15) and TG Condition (trust/distrust) as fixed effects, and participants as random effects of varying intercepts. Namely, the LME-model shows that there is a difference in the economic behavior between the two TG Conditions when each group is playing with Angela (by increasing or decreasing offers throughout the game to correspond to the behavior of Angela i.e., adjusting to his behavior within the course of the game). In contrast, the LW-group managed to adjust their behavior according to the trustworthy/untrustworthy version of Berlusconi, while the RW-group did not.

Comment No. 3

“As a side point, I was also wondering if participants found the trustworthy and untrustworthy interactions with Berlusconi (and to a lesser extent, Angela) believable. I don’t think this is critical to the manuscript, but is there any way to assess whether the trustworthy vs. untrustworthy conditions reflect subjects’ preexisting perceptions of Berlusconi?”

We are grateful for this comment as it enables us to highlight those segments of the manuscript that are specifically related to the mentioned concerns, and to present them in a more detailed and concise manner.

Firstly, we would like to underline that participants’ distrust in the cover story was the main exclusion criteria in the analysis of their economic behavior. Therefore, we exempted three subjects who did not believe in the cover story, and included a selection of 118 subject in the final analysis of the economic behavior (as already reported on page 3 of the manuscript, under the section “Materials and Methods: Participants”). Participants’ trust in the cover story was assessed with a carefully designed manipulation-check protocol in order to extract their response without disclosing/suggesting the aim of the study in advance. Therefore it started with the following indirect questions: “According to you what is the scope of the experiment?” (to which participants provided written explanations) and “Was the mathematical model predictive of Trustee’s behavior?” (to which they responded by using a 0-100 VAS scale). The participants who indicated predictability of the mathematical model lower than 30% (the cutoff value), were again asked to provide written justification for their reasoning. At the end, participants whose summary answers on all three questions implied that they guessed the real aim of the experiment, were directly asked whether they believe in the cover story. A negative response was taken as the main reason for exclusion of the participant from the analysis of the economic behavior. The participants who were included (i.e., believed in the story about “the mathematical model”), were surveyed regarding the reliability of “their partner” in the game (i.e., they provided post-experimental evaluation of how un/trustworthy was the Trustee during the game, on a 0-100 VAS scale).

The post-experimental ratings of Trustee's economic behavior were analyzed and confirmed that participants were explicitly aware whether he behaved in trustworthy/untrustworthy manner, regardless of who the partner was (SB or PA), and to which group he/she belonged (LW or RW). Namely, these ratings on Trustee's Perceived Trustworthiness (0-100 VAS) were analyzed in a 2 x 2 x 2 ANOVA, with the Group (LW/RW), TG Condition (trust/distrust) and Trustee (SB/PA) as between-subjects factors. This analysis did not result in a significant triple interaction ($F(1,109) = .88, p = .351$), showing that the two groups did not differ significantly in the explicit trustworthiness ratings of their respective partners, and for the respective TG conditions. Instead, we did find a significant main effect of the TG condition $F(1, 109) = 47.13, p < .001$, showing that both groups attributed higher/lower trustworthiness ratings to the respective version of the game. This suggests that, at least at the explicit level, participants' evaluation of their partners' behavior is not based on preexisting perception about his trustworthiness.

Finally, to directly address the reviewer's concern, we ran a simple linear regression analysis where the scores which participants attributed to Berlusconi's trustworthiness prior to the experiment (on a 5-point Likert scale) were used as a predictor variable, while their post-experimental ratings of Berlusconi's trustworthy/untrustworthy behavior throughout the game (on a 0-100 VAS scale) were employed as a dependent variable. The results were not statistically significant ($F(1, 58) = .62, p = .433$), meaning that there was no linear relationship between the two variables. Hence, the first measure which was taken as an indicator of a preexisting bias regarding Berlusconi's trustworthiness did not affect the perception of Berlusconi's trustworthy/untrustworthy behavior during the game i.e., the explicit evaluation was unbiased and realistic.

Comment No.4

“Could the measures used for the Explicit Political Orientation questionnaire be provided in the text? Considering this is the IV used in the primary analyses, I was hoping to understand how ideology was measured – for example, was it a single self-placement item, a composite measure of different ideological dimensions, or a composite measure of positions on a variety of policies?”

We appreciate this thoughtful comment, which invites us to provide detailed explanations, and creates space for additional procedural descriptions. In line with reviewer’s presumptions, participants’ political orientation was measured via direct question i.e., by asking participants to self-place on a 7-point Likert-type item (1=“extremely left”; 2=“left”; 3=“center-left”; 4=“center-right”; 5=“right”; 6=“extremely right”, 7=“apolitical”). According to the responses, participants were then split in two groups, with the LW-group consisting of voters who assigned scores from 1 to 3 on the self-placement scale, while the RW-group of voters who self-assigned scores ranging from 4 to 6. Those who rated themselves as apolitical (i.e., selected “7”) were excluded from the experiment. We also introduced several control questions asking participants to specify the voted party at the three last consecutive elections (in order to make sure that selected participants had clear idea regarding their own political orientation), and to specify to whom and how often they talk about politics (in order to have extra measure about participants’ interest in politics).

In order to make this point more clear, we inserted a new paragraph under the “Material and Methods: Procedure” section of the manuscript which now reads as follows: “Participants’ self-reported ideology” (page 3, lines 49-58).

Comment No.5

“I think the model using the social conservatism factor is interesting and illuminates potential mechanisms by which the observed ideological differences may emerge. Would it be possible to include a similar model using the economic liberalism factor, perhaps in the SI? It seems a lost opportunity to measure these constructs, which likely also play a role in the economic behavior in this context, but not to explore them.”

We share reviewer’s appreciation for the additional ideological indicators (which can be extracted from measures like MF, RWA, ESJ, SDO, SVO).

As regards reviewer’s specific suggestion to address the Economic Liberalism in addition to the Social Conservatism factor, as mentioned on page 5 of the manuscript, the principal component analysis did reveal a two-factor structure. The Economic Liberalism mainly loaded by MFQ Fairness, MFQ Harm and SDO, while the Social Conservatism by MFQ Purity, MFQ Authority, MFQ Loyalty and RWA. The dimensions emerged with the varimax orthogonal rotation i.e., they were considered as independent to each other and therefore they could have different and autonomous contributions in explaining of participants’ behavior during the TG. As asked by this reviewer, we repeated the same analysis on both factors.

Specifically, by following the same procedure adopted for Social Conservatism factor analysis, we applied several models of increasing complexity including the Economic Liberalism. The full LME model had the following fixed effects: Trial No (1-15), TG Condition (trust/distrust), Trustee (SB/PA) and Economic Liberalism Scores (with interaction term), while the participants were always treated as random effects with varying intercepts. The results revealed that, differently from Social Conservatism, the full-interaction model was not marked by AIC drop, or by significant change in the LR (as compared to its nested model).

In light of these results and in line with our preliminary hypothesis, we think that only Social Conservatism provided meaningful explanations about participant's economic behavior over the entire course of the economic exchange. Upon reviewer's request, a supplemental figure is added to this response, representing a heat map model (Fig. 1) of participants' economic behavior in function of their Economic Liberalism scores. As shown on the map, the Economic liberalism did not add meaningful insights on participants' economic behavior throughout the game.

Fig. 1 Heat map model of the economic behavior of participants as a function of their level of Economic Liberalism. The panels on the left indicate participants' economic behavior during the Trust Game with Piero Angela (PA), while those on the right indicate participants' economic behavior throughout the Trust Game with Silvio Berlusconi (SB). The upper panels indicate the change of participants' behavior in response to a trustworthy game partner, while the lower panels display the participants' adjustment to an untrustworthy game partner. The color-grading provides coarse-grained orientation on the probability to invest a certain amount: red indicates low, while yellow color high probability to invest. The legend provides fine-grained insight into the numerical gradient of change in investment.

Comment No. 6

“Finally, it would be helpful to include some discussion about the political implications of these results. It’s not totally clear to me how one-to-one interactions and exchanges with political leaders are realistic, but perhaps the authors could consider other economic scenarios/games in which the investments are explicitly framed as donations and trustees’ behavior is linked to some policy outcomes, for example.”

In the reply to Comment No. 3 by this reviewer, we provided additional explanations and empirical evidence supporting the idea that our participants genuinely believed in our cover story throughout the course of one-to-one TG interactions, so we hope that the reviewer is now convinced in this point. We find his/her concluding remarks (about the wider political implications from our results) to be in line with our reasoning which was also stated in the manuscript. Those considerations were the driving force behind our decisions to specifically choose the trust economic game over the other economic scenarios/games.

Below we will provide the rationale for such choice and indicate the paragraph of the paper where we discuss it:

- In the introductory part (page 2) of the paper we motivate the choice of using a TG scenario: “Leaders use extended media presence in order to exert influence over potential voters, by means of projecting admirable personality traits, portraying them as desirable candidates. Hence their roles (e.g., leaders as trustworthy and voters as trusting subjects) and the incentives (e.g., economic offers) may be reflected in the experimental design of the Trust Game (TG).”*
- We also considered the fact that TG realistically reflects the relationship between the followers and the political leaders, since each voter is also a tax-payer (i.e., trustor) while the political leaders with governmental experience are decision-makers (i.e., trustees) who can choose to spend the budget for societal improvements (thus returning the trust and behaving trustworthy), or for their own benefits (thus behaving in an untrustworthy manner).*

SAPIENZA
UNIVERSITÀ DI ROMA

- Moreover, we decided to select a TG design because our study pioneered in using a cover story about a supposed mathematical model (claiming “a realistic representation” of a political leader) which allowed for one-on-one interaction between the leader and the voter. A donation paradigm would not return in a reciprocation towards the electors from the political leader (or the famous character). Finally, a different paradigm would not allow us to test the plasticity of trusting an in-group/out-group political leader (with respect to a famous non political polarized character) which was another important aim of the present study. Hence, we decided to go with a well-established experimental design resulting in findings that would be clear for analysis, interpretation, and comparison with similar studies.

REVIEWER No.2

Comment No. 1

“I am not entirely sure what to make of this manuscript. It is interesting, but to me it seems more like a promising pilot study than a definitive study. I am not entirely sure what it means, psychologically (to research participants) to play a trust game with Berlusconi’s ‘algorithm’ but it does raise some intriguing questions.”

Firstly, we would like to thank this reviewer for the efforts dedicated in the revision of our manuscript. We found his/her comments very useful, since they opened a floor for fruitful discussion, inviting interesting considerations and additional elaborations along the way.

As to the comment that the presented research may seem like a promising “pilot study”, we believe that given its pioneering nature, our study might pave a way towards replications on large populations (testing, for example the postulated hypotheses in various geopolitical contexts and with different political leaders). Also, supported by multiple analyses, we strongly believe this initial study provides important clues on the processes underlying the trust of in-group/out-group voters toward their political leaders.

We agree with reviewer’s comment that an interaction with a living person rather than an “algorithm” may be preferable, and we are aware that participants’ perception of a model of a person, might not equal with their perception of that very person. At the same time, we expect that the reviewer will agree with us upon the fact, that in the real life, a real inclusion of a famous leader in a role of a trustee, is hardly achievable.

Therefore, we introduced a cover story which was considered plausible according to participants’ ratings during the manipulation-check protocol. The cover story informed the participants about “a mathematical model” which calculates trustee’s responses to trustor’s monetary offers, based on a psychological/ideological assessment of the real trustor (from their responses to the questionnaires),

and on the basis of match/mismatch with the trustee's profile. Here we want to highlight the fact that the cover story enabled an ecological reference, produced credence and have been massively used in the psychological research (Dickson, 2011)¹.

Because we relied on the cover story in order to induce a sense that the economic interactions are realistic in their nature, we took participants' distrust in the cover story as the main criteria for their exclusion upon the analysis of the economic behavior. Therefore, we exempted three subjects who did not believe in the cover story, and performed the final analysis on a selection of 118 subjects. As mentioned in the response to the first reviewer, participants' trust in the cover story was assessed via carefully designed manipulation-check protocol, to extract their response without disclosing/suggesting the aim of the study in advance. Namely, it started with the following indirect questions: "According to you what is the scope of the experiment?" (to which participants provided written explanations) and "Was the mathematical model predictive of Trustee's behavior?" (to which they responded by using a 0-100 VAS scale). The participants who indicated predictability of the mathematical model lower than 30% (the cutoff value), were again asked to provide written justification for their reasoning. At the end, participants whose summary answers on all three questions implied that they guessed the real aim of the experiment, were directly asked whether they believe in the cover story. A negative response was taken as the main reason for exclusion of the participant from the analysis of the economic behavior. The participants who were included (i.e., believed in the story about "the mathematical model"), were surveyed regarding the reliability of "their partner" in the game (i.e., they provided post-experimental evaluation of how un/trustworthy was the Trustee during the game, on a 0-100 VAS scale).

On a final note, while an "algorithm" certainly can't substitute a real person, the tangibility of our results regarding the adaptive behavior of LW-voters according to all our scenarios (trustworthy and untrustworthy versions of Angela/Berlusconi) and of the RW-voters according to a part of all our scenarios (trustworthy and untrustworthy versions of Angela) suggests that participants were indeed genuinely involved in the economic game.

Comment No. 2

“I am not convinced that such a strange situation really says much about “voter-leader interactions” (p. 7).”

A similar concern was also raised in the last comment of the first reviewer, who expressed uncertainty as to “how one-to-one interactions and exchanges with political leaders are realistic”. So, we will refer to our previous response in order to address this issue:

- In the introductory part (page 2) of the paper we explain our motif behind the choice of the Trust Game: “Leaders use extended media presence in order to exert influence over potential voters, by means of projecting admirable personality traits, portraying them as desirable candidates. Hence their roles (e.g., leaders as trustworthy and voters as trusting subjects) and the incentives (e.g., economic offers) may be reflected in the experimental design of the Trust Game (TG).”

- We also considered the fact that TG captures the core of the relationship between the followers and the political leaders, since each voter is also a tax-payer (i.e., trustor) while the political leaders with governmental experience are decision-makers (i.e., trustees) who can choose to spend the budget for societal improvements (thus returning the trust and behaving trustworthy), or for their own benefits (thus behaving in an untrustworthy manner).

Hence, a computerized Trust Game seemed to us a well-validated experimental paradigm for capturing complex voter-leader relationships, and a plausible approach to investigate behavioral correlates of leader-voter trustworthiness.

Comment No. 3

“My main concern, however, is that the sample is too small to support meaningful conclusions, and it is potentially problematic that there was no left-wing equivalent to Berlusconi included in the design of the experiment.

The final sample was $N = 118$ (p. 4), and (before exclusions) there were 58 rightists and 63 leftists altogether. There were two between-participants experimental conditions, which means that there were roughly 24-32 (or less, after exclusions) participants in each condition. I am afraid that these numbers are just too small to warrant the drawing of conclusions about rightists and leftists in Italy. I have not performed any power analyses, but I suspect that the study is dramatically under-powered.”

Firstly, we thank the reviewer for the raised issues regarding the sample size, and the generalizability of our results, since they provide an opportunity to elaborate on the additional analyses in support of our findings. Before proceeding with the supporting evidence, we would like to clarify that the final sample constituted of 118 participants, since only 3 participants were excluded from the analysis on economic behavior due to their distrust in the cover story.

We did use a frequentist approach to investigate our hypothesis, but at the same time we were very careful to employ different types of statistical approaches (starting with ANOVA's where we considered the means of each group and condition, ending with mixed models analysis where we considered all the data points we collected in the experiment). As suggested by the reviewer, the p-value depends on the number of participants, so we decided to back-up our findings with a Bayesian inferential approach as a way of solving possible problems with the power-analysis.

To this purpose, we performed the Bayesian regression models with Stan package (Bürkner, 2018). For our analysis, we set weakly informative priors on the estimated parameters as explained in the text. Then, posterior credible intervals (95% PCI) to quantify our beliefs on each parameter. The degree of relative evidence in favor of the Alternative Hypothesis ($H_1: \beta \neq 0$) compared to the Null Hypothesis ($H_0: \beta = 0$) was computed through the Dickey-Savage density ratio, by comparing the

probability density of H_0 under the prior distribution vs. the posterior distribution ($B F_{10}$). For the sake of simplicity, when the data support the null, we report the probability density of H_0 under the posterior distribution vs. the prior distribution ($B F_{01}$), which measures the relative evidence in favor of the null (vs. the alternative). To describe the amount of evidence, we use the labels proposed by Raftery (1995), as reported in Jarosz and Wiley (2014).

In the following section we refer to the main findings of a key-performed analysis:

To explore the overall dynamics of trust we employed linear mixed effects modeling (LME) in brms. This approach allowed us to thoroughly investigate the relationship between the economic behavior indexed by the amount of investment offered by participants at each trial and their self-reported political orientation (LW/RW). Therefore, we specified several models of increasing complexity with the simplest as a random-intercept model, allowing only between-subject variance in the average investment. The full model had the following fixed effects: Trial No (1-15), Group (LW/RW), TG Condition (trust/distrust) and Trustee (SB/PA), along with all the possible interactions across the variables. Participants were always treated as random effects with varying intercepts. The model selection relied on the Watanabe-Akaike Information Criterion (WAIC). The results revealed that the full-interaction model was marked by both WAIC drop from 8118.62 (model with the main effect of Trial No, TG Condition and their interaction) to 8115.63 (full model) ($\Delta WAIC = -2.99$). Thus, the full-interaction model was selected as a best fit, showing that the economic behavior of the trustor changed as a function of the interaction between trustee's and own political orientation, and their mutual economic exchange. The changes in trustor's behavior unfolded throughout 15 exchanges, hinting at a considerable short-term plasticity of trust. Crucially, we found positive evidence in favor of the four-way interaction ($B F_{10} = 17.06$).

Comment No.4

“Furthermore, the failure to include a left-wing equivalent to Berlusconi (like Romano Prodi) means that we cannot really tell whether there was an ideological asymmetry in terms of trust behavior. It is possible that leftist participants would persist in trusting Prodi, but it is also possible that they would not. Until this question is answered, it is not clear to me what we can conclude from this study.”

We share the reviewer’s interest in considering the possibilities for ideological asymmetry. However, we suggest that it might be a promising direction for exploration in countries with strong LW-leaders. As for our study, we want to highlight the fact that we referred about the behavior of the RW-group towards an in-group leader, without the aim to draw any conclusions regarding the behavior of the LW-group towards a hypothetical in-group leader. In fact, we agree with the reviewer that, at this point, it remains unclear whether an ideological asymmetry will emerge, or the results will be mirrored in both groups.

We added a paragraph in the discussion of the revised manuscript to highlight the point (page 9, lines 17-21).

In addition, we will point to segments of our previous response (to the first reviewer) where we explain our caution in drawing parallels upon shaping of the paper:

- The original title “Right-wing electors blindly invest in their leader in the economic trust game” highlights the most prominent result related solely to the behavior of the RW-group i.e., the fact that they displayed constant pattern of behavior towards their leader regardless of the trust/distrust version of the game. However, the title does not suggest that such behavior is juxtaposed (or in any way comparable) to the behavior of LW-group, had they been placed in a similar situation.

- In the discussion we reason that it might be “intuitive to expect that one will act irrationally when s/he is ideologically biased. In fact, RW-voters behaved in line with this reasoning: they did not adjust to trustworthy/untrustworthy behavior of their in-group leader, i.e., Berlusconi”. However, we also argue that it might be “counter-intuitive to expect that one will act rationally when s/he is ideologically biased. In our experiment, LW-voters behaved rationally in spite of this reasoning: they adjusted their economic behavior to the trustworthy/untrustworthy behavior of the out-group leader Berlusconi”. Here again, we are careful to interpret our findings according to the collected

evidence. We did not extend the comments to a hypothetical situation of LW-group playing with an in-group leader. In fact, we go on to speculate that such behavior might be attributed to “the fact that they were not playing with an in-group leader” which, in a way, is opposite to the claim about ideological asymmetry.

Finally, we motivated our decision to refrain from including a left-wing politician (with special explanation about the suggested Romano Prodi). In the concluding remarks of the paper, we clearly affirm that “in Italy at present, we could not recognize a single LW-leader with same level of popularity as Berlusconi”. According to us, the crucial dimensions for control in the choice of a renowned/popular LW-leader (i.e., Berlusconi’s equivalent) are the following:

- Long-term political presence and governmental leadership experience (for example, a leader of a coalition/party, or a prime minister of the country);*
- Persisting popularity via continuous public exposure and media appearance;*
- Clear perception of leader’s political orientation i.e., a person with unambiguous political categorization.*

As we already report in our previous response (to the first reviewer), in Italy we could not find a LW-politician who embodies all required qualities. Indeed, all the selected candidates differ from Berlusconi in something more than ideology. For instance, the Italian Prime Minister at the time of the experiment, the much younger LW-leader Matteo Renzi, was not elected by citizens but chosen by the President of Republic with mandate to form a technical administration. Also he lacked long-term political presence and media appearance. On the other hand, the career of the LW-politician Romano Prodi, is indeed marked with long-term political presence (Prime Minister in the periods of 1996-1998 and 2006-2008) which is comparable to Berlusconi’s leadership experience (Prime Minister in the periods of 1994-1995, 2001-2006 and 2008-2011), but he lacks ongoing public exposure and media appearance (since his decade-long retirement from politics).

We added a paragraph in the discussion of the revised manuscript to highlight the point (page 9, lines 22-40).

In addition, the politicians who are currently popular are not as clearly recognized or politically categorized, as Berlusconi. This claim is supported by data from a recent experiment (conducted in 2018, for a separate study). Namely, in an independent sample of 120 RW/LW Italian voters, we tested their level of recognition and political categorization of the most prominent Italian politicians belonging to the LW-coalition, RW-coalition, or the party called “Movimento Cinque Stelle”. Specifically, participants were asked to observe the faces of 30 liberals, 28 conservatives, and 9 popular politicians from the “Movimento Cinque Stelle”. After the exposure, they were asked to attribute a name to each face (recognition) and rate the politicians’ ideology (political categorization). This is similar to the evaluation performed automatically by the students participating in the trust economic game, since we also included a picture of the trustee (which requires prior recognition) and statistical analysis of these data confirm that the only politician who was unanimously recognized and equally categorized as conservative leader by 100% of the respondents, was indeed Berlusconi. On the other hand, the LW-politicians who were either unanimously recognized (like Matteo Renzi), or politically categorized (like Nichi Vendola) by all respondents, did not receive full recognition/categorization respectively. Also, we would like to highlight the fact that these results are collected in 2018, when Renzi was already recognized as a former Prime Minister (period of 2014-2016), while Vendola became prominent LW-politician (former president of Regione Puglia, 2005-2010, and leader of the national left-wing party “Sinistra Ecologia e Libertà, 2010-2016). Since we couldn’t rely entirely on the prominent LW-politicians we also analyzed the political recognition/categorization for the representatives from “Movimento Cinque Stelle”. As expected, their most prominent candidates had very high recognition, but very ambiguous political categorization.

In summary, we believe that the political settings in Italy at the time of data collection did not allow us to find any LW-match to Berlusconi. However, we strongly support the idea of replicating the study in a different geopolitical context and recommend “that future studies from countries where a left-wing equivalent (with matching years of political presence, leadership experience, and media appearance) exists, can help to address the question about the power of the leader on both LW- and RW-voters”.

Comment No. 5

“There are other concerns I have as well. If I am reading things correctly, the Berlusconi condition was run many months before the control condition (Angela). This means that there is a confound between time (or season) and interaction partner.”

As rightly noted by the reviewer, the study was conducted in two separate experiments for two consecutive periods during 2014 (Exp 1: January-July and Exp 2: October-December). The main experiment (Exp 1) utilized Silvio Berlusconi, former Prime Minister of Italy and leader of the center-right parties' coalition, while the control experiment (Exp 2) utilized Piero Angela, a famous TV host. However, we believe that there is no confound between time (season) and the interaction partner based on the following reasons:

- We decided to choose Berlusconi as our main stimulus for the first experiment and to conduct Exp1 in a single batch, specifically for the purpose of avoiding the confound of time on the public perception of Berlusconi. Namely, we wanted to avoid the possibility that unexpected political developments might affect the public image of Berlusconi. Indeed, a political trust-index may quickly fluctuate due to the constant political engagements and public affairs. That's why his trust-index was closely monitored with bimonthly reports, obtained by “Istituto Piepoli” (an agency specialized in marketing and opinion research) revealing 31%, 31%, and 34% of the total voting-pool, for the months of January, April, and June (i.e., remaining fairly stable throughout the testing period).

- On the other hand, we did not expect considerable fluctuation in the public perception of Angela, and we were not restrained by the period/season for the Exp2. He is a famous non-politician with long-term media presence, but stable public appearance. This was confirmed in both, the pilot study and the experimental study, where participants on both sides did not differ in their perception of Angela and considered him as a relatively neutral character.

SAPIENZA
UNIVERSITÀ DI ROMA

At the end we want to remind that we utilized between-subjects design, employing separate participants for each Group (LW/RW), Trustee (SB/PA) and Condition (trust/distrust). Hence, no subject was involved twice in a same procedure (to play with separate Trustees or different Conditions) and the possibility for the confound between period, interaction partner and condition was avoided altogether. Moreover, we are not familiar with possible confounding effects represented by seasons over the political interaction partner.

Comment No. 6

“I also have some questions about the results (and how to interpret them). Why were there no differences between leftists and rightists in terms of (a) how much they invested in Berlusconi vs. Angela (p. 5), and (b) explicit trustworthiness ratings of Berlusconi vs. Angela (p. 6)? I find these null results to be surprising, and they make interpretation of the behavioral results more difficult, in my judgment.”

This is a thoughtful remark, and we are grateful for the possibility to offer our interpretation about the pointed findings.

We agree with the reviewer that the lack of significant double interaction between the Group (LW/RW) and the Trustee (SB/PA) in the first investments may appear counter-intuitive. Two non-mutually exclusive explanations can be offered for the initial lack of bias. The first is that although the participants believed in the cover story about the “mathematical model”, they started out by testing the game and become gradually more involved in the interaction with their partner as the game progressed. The second possible explanation of the model is that at the time of the experiment Berlusconi was no longer in a position of highest power and his ancient charisma might have been reinforced as the interaction developed.

The lack of the significant triple interaction between the Group (LW/RW), TG Condition (trust/distrust) and Trustee (SB/PA) on the perceived trustworthiness ratings, showed that both groups attributed higher/lower trustworthiness ratings to the respective version of the game. So, it suggests that the participants were really involved in the interaction with their partners on the overall, since they were consequently able to make explicit distinction between Trustee’s trustworthy and untrustworthy behavior. We find this result to be in line with the existing literature on group processes (Avenanti et al., 2010) and studies on differences between implicit and explicit measures in partisans (Galdi, Arcuri & Gawronski, 2008; Nevid & McClelland, 2010). Also we believe that it provides additional insight about the ideological bias regarding the trust in a political leader, so we decided to include it in our report.

Comment No. 7

“In terms of theoretical issues, I don’t think that this paradigm can tell us anything about whether liberals and conservatives differ in terms of cognitive rigidity (p. 7).

We thank the reviewer for his comment, because it opens space for elaboration on the wealth of evidence supporting our discussion points.

Regarding the issue of cognitive rigidity, in a study by Amodio et al (2007), the authors provide neurophysiological support for the claim that “stronger conservatism (versus liberalism) is associated with less neurocognitive sensitivity to response to conflicts”. This is in line with other behavioral studies by Jost et al (2003), and Kruglanski et al (2006), where conservatives exhibit overall tendency for habituation to fixed or repetitive responses, with lack of sensitivity for conflicting responses.

We believe that in our study, such dispositional inclination in the RW-group would have produced a lack of adjustment to both experiments, regardless of the interaction partner. At a minimum, we could have expected lack of adjustment to Angela’s untrustworthy behavior. However, the fact that the RW-group demonstrated lack of adjustment to (un)trustworthy behavior of their in-group leader (but not the control character), implies that when it comes to the ideological context more powerful forces might be at play. We now hope that the parallelism is made clearer.

In order to stress the point more explicitly, we expanded the paragraph accordingly (please refer to page 7, lines 55-59 and page, 8 lines 6-11).

Comment No. 8

“I also don’t know what it has to do with moral foundations theory, except to the extent that the behavior is tapping into authoritarianism. I am unsure about the connection to gaze-attraction as well.”

With regards to the ideology, we like to elaborate on the relationship between moral values and the authoritarian attitudes. So far, Kugler et al (2012), have provided direct evidence that “liberal-conservative differences in moral intuitions are statistically mediated by authoritarianism and social dominance orientation, so that conservatives’ greater valuation of in-group, authority and purity concerns is attributable to higher levels of authoritarianism”. In support of this finding, our PCA analysis revealed that the first factor was mainly loaded by the following variables: RWA social attitudes (.78), and purity/sanctity (.85), respect for authority (.83) and in-group loyalty (.70) as moral values. So, we have an additional reason to believe that an interaction with a political leader is marked by top-down regulation, in which social attitudes and moral values affect the decision-making processes to trust or not to trust a leader.

We expanded the paragraph accordingly to include more extensive explanations (please refer to page 8, lines 18-32).

Finally, the studies by Luizza et al., (2011) and Porciello et al., (2016) involving gaze-following behavior of RW- and LW- Italian voters are referenced as relevant for our case, since they provide behavioral and physiological evidence about the capability of a conservative leader to exert influence over his followers (as compared to leftwingers). Moreover, by highlighting the behavioral (i.e., oculomotor) changes of RW-participants in response to the authority of their leader (again represented by the same person i.e., Silvio Berlusconi), these studies provide evidence that the observed difference between conservatives and liberals might be due to their heightened sensitivity toward the authority of the leader (Altemeyer 1996), rather than being more loyal to their own group (Haidt and Graham, 2007; Graham et al., 2009). According to us, this is in line with the results reported in our manuscript on the economic behavior towards the conservative leader and the ideological dispositions of the voters.

Appendix B

SAPIENZA
UNIVERSITÀ DI ROMA

To:

Dr. Antonia Hamilton (Subject Editor)

Dr. Molly Crockett (Associate Editor)
Dr. Andrew Dunn (Senior Publishing Editor)
at *Royal Society Open Science*

Date/Place:

1 May 2019, Rome, IT

SUBJECT:

Re-submission of the research article RSOS-182023 entitled “Bound to the group and blinded by the leader: ideological leader-follower dynamics in a trust economic game” (originally titled “In leaders we trust: right-wing electors blindly invest in their political leader in an economic trust game”)

Dear Editors at *Royal Society Open Science*

Dr. Antonia Hamilton and Dr. Molly Crockett,

We would like to express our gratitude to the Team of Editors and the Team of Reviewers at the *Royal Society Open Science*, for the expert handling of the manuscript by Gjoneska, Liuzza, Porciello, Caprara and Aglioti, originally entitled “In leaders we trust: right-wing electors blindly invest in their political leader in an economic trust game”, and (following a substantial revision) updated to “**Bound to the group and blinded by the leader: Ideological leader-follower dynamics in a trust economic game**”. Also, we thank you for allowing us to revise the manuscript along lines that proved very helpful in our striving to improve it.

We hereby confirm that we did follow the main recommendation by the Associate Editor “*to address a number of substantial matters*” and we did our best to “*ensure that we resolved the major concerns by the reviewers*”. This is evident in the *considerable interventions at all crucial points of the manuscript* (i.e., the title, the abstract, the introduction, the discussion and conclusions). They are introduced for the purpose of *shifting the focus of the study, from highlighting the right-wing/left-wing differences when relating to a political leader* (as was the case in the first version), *to highlighting the group-differences when relating to in-group/out-group political leader* (as is the

case in the updated version). In this way, we have moved away from the claims which are yet to be supported by other studies, towards the claims that are clearly evidenced in our results.

We believe that *the new material is framed more precisely and concisely, in a more constrained and cautious manner*, thus it is in good accordance to the main comments and concerns, suggestions and recommendations, from the editor and the reviewers.

In addition, please refer to the rebuttal letter containing our response (which is marked in **bold**) to the reviewers' comments (which are cited in "quotes", marked with *bold and italic*, and **highlighted in blue**). The revised segments of the manuscript are also included in the response (and cited in "quotes", plus **highlighted in yellow**).

We do hope that you and the reviewers will find the response to be clear and that the manuscript is now suitable for publication in the *Royal Society Open Science*.

On behalf of all authors we thank you.

Sincerely yours,

Salvatore M. Aglioti and Biljana Gjoneska

Social and Cognitive Neuroscience Laboratory

Department of Psychology

School of Medicine and Psychology

Sapienza University of Rome

Postal Address: Via dei Marsi 78, 00185, Roma, Italy

Web Address: <https://agliotilab.org/>

Correspondence

Biljana Gjoneska: biljanagjoneska@manu.edu.mk

Salvatore M. Aglioti: salvatoremaria.aglioti@uniroma1.it

SAPIENZA
UNIVERSITÀ DI ROMA

RESPONSE TO REVIEWERS

Introductory remarks:

The following response contains a comprehensive explanation for each intervention, made at crucial points of the manuscript. All revisions have been implemented for the purpose of providing information which is strictly related to specific testing conditions i.e., when supporting/opposing group of voters (in the role of trustors), are playing with an in-group/out-group RW-leader or a famous non-politician (in the role of trustees) an iterated version of the trust economic game. Hence, the entire material is restructured in order to refrain from extending conclusions to conditions that are not included in our experiment (e.g., playing a game with a LW-leader in the role of a trustee). In this way, the study manages to compensate for the lack of suggested condition, because such condition could not be introduced under the current circumstances due to: the temporal constraints (i.e., short time-window for the replication of the experiment), political constraints (i.e., changed political climate, compared to the period when the experiment was conducted), and the methodological constraints (i.e., the unavailability of same voters, or a pool of voters with similar specifics as the ones in our initial experiment). As a result, the rebuttal letter will focus on answering the suggestions/comments from reviewers which are still applicable for the new version of the restructured manuscript.

Legend:

In the following lines we will provide precise order of all interventions in the revised manuscript, along with exhaustive explanations for each modification. Our response will be marked **in bold**, while the revised excerpts from the manuscript will be copied, quoted and **highlighted in yellow**. Also, the addressed comments from the reviewers will be quoted and **highlighted in blue**, and marked with ***bold and italic*** across the text.

I. Adaptation of the TITLE

The new version of the title reads **“BOUND TO THE GROUP AND BLINDED BY THE LEADER: IDEOLOGICAL LEADER-FOLLOWER DYNAMICS IN A TRUST ECONOMIC GAME”**

This modification is introduced, in direct response to the comments from Reviewer 1. Also, it is in line with his/her suggestions (made on both rounds of the revision) which are cited as follows:

- “At minimum, if a left-wing politician condition cannot be included, I think the language about the ideological differences should be softened substantially” (First revision)

- “At minimum, I think the language about the design and the parameters that are actually being tested would be better served with greater constraint and precision” (Second revision)

The rationale for such intervention is structured around three key-points and explained as follows:

1. The whole title is framed in a more precise manner, in order to directly address the concrete results from our study. Specifically, the revised version of title (i.e., “bound to the group and blinded by the leader”), pertains to the factual and tangible evidence on moral values and social attitudes of voters (extracted with Principal Component Analysis from the corresponding questionnaires), in relation to their economic behavior (driven by their dis/trust towards an in-group/out-group leader). Namely, the analysis has shown that the observed changes in voters’ trust economic behavior (or lack of any thereof), were due to their inclination for in-group loyalty and respect for the authority of the in-group leader, so this conclusion is synthesized in the first part of the title. In addition, the subtitle (i.e., “ideological leader-follower dynamics in a trust economic game”) emphasizes the setting and the circumstances under which the results emerged. Namely, it serves to provide information about the methodology (i.e., trust economic game) and the experimental design (i.e., game designed to investigate ideological leader-follower dynamics), rather than speculations about the causality of the results (as was the case with the original title).

2. The new title is framed in a more constrained manner in order to avoid any inferred messages, potential interpretations, possible implications, or any hints whatsoever, which did not stem from our results. Namely, the new title restrains from mentioning (or hinting at) potential ideological asymmetry between the RW/LW group of voters, since this was not directly tested in our experiment, and it is not supported by our data.

3. The new title is framed in a more general manner, in order to focus on the more universal intergroup processes and in-group biases, rather than what might turn to be a single-case scenario (if our pioneering study is not replicated in future studies utilizing different controls, or other political contexts).

II. Adaptations in the ABSTRACT

The new version includes the following modifications of the abstract (highlighted in yellow):

“Results revealed that depending on the group, voters either relied on the situation and adjusted to the behavior of the out-group leader (in our case left-wing voters), or on their disposition for group-loyalty with respect for authority, thus failing to adjust to the behavior of the in-group leader (in our case right-wing voters). Our findings suggest that: a) complex voter-leader relations in politics, are reflected in the simple trustor-trustee financial interactions from behavioral economics and b) being bound to one’s group and one’s leader may affect the trust economic decisions of the followers.”

The modifications are introduced as a result of the insightful observations by Reviewer 1, cited as follows:

“There are many points in the manuscript that suggest an ideological asymmetry that I just don’t think is supported by the data. For example, in the abstract, it is stated that ‘results revealed that left-wing voters relied on the situation (trustee’s behavior), while right-wing voters did not’. This suggests an asymmetry between left wingers and right wingers, and it is not supported by the data—both left wingers and right wingers updated their investment behavior in response to the non-politician.”

The rationale for the modifications is structured around two key-points, and explained as follows:

1. The focus of the study is shifted, from highlighting the RW/LW differences when relating to a political leader (as was the case in the first version), to highlighting group-differences when relating to in-group/out-group political leader (as is the case in the updated version of the abstract). In such a way, we have implemented the reviewer’s recommendation and moved away from the potential claim about an emerging ideological asymmetry (since this remains to be supported with additional studies), towards the claim about being biased when playing with one’s own leader (since this is evidenced in

the results). Hence, the language about the ideological differences is softened considerably, as was originally suggested by Reviewer 1.

2. The focus of the study is sharpened, and better structured to delineate a) the investigation of the general dynamics in a leader-follower trust economic game, and b) the investigation on the mentioned group-differences when relating to in-group/out-group political leader. This was done in synchrony with the adaptations in the title.

III. Adaptations in the INTRODUCTION

The specific additions in the introduction (highlighted in yellow), are cited as follows:

“Furthermore, voters who score higher on binding moral values tend to support social attitudes that lead to in-group favoritism, with promotion of social hierarchy and inequality [11, 12]. This in turn, may lead to biased perception about trustworthiness of in-group members (especially if they are leaders placed high on the hierarchical ladder), and may produce bias in variety of social behaviors. The bias may also become evident on daily bases, and reflected in the daily decisions to trust like-minded partisans not only on political issues but also on unrelated, non-political matters (the so-called “epistemic spillover”) [13]. For example, Marks et al., found that the similarity with one’s political views affects one’s ability to make accurate assessment about their fellow’s expertise in the domain of geometric shapes. Moreover, trusting in-group politicians (especially when they are powerful leaders) can happen even when this implies spreading disinformation [14].” (page 2, lines 24-32)

The additional information is inserted in response to the insightful observations and useful suggestion by Reviewer 1, cited as follows:

“Much of the text suggests that the authors are exploring a balanced design (e.g., on p. 2: ‘we were able to capture the initial trust of LW vs. RW participants towards an in-group vs. out-group leader, hence potential positive vs. negative bias toward him’), but the design simply tests an in-group politician for RW participants and an out-group politician for LW participants. At minimum, I think the language about the design and the parameters that are actually being tested would be better served with greater constraint and precision”.

Specifically, the introductory segment is inserted for the purpose of providing context to our findings (which are described later in the manuscript). Namely, it provides information on two key-concepts:

1. The so-called “epistemic spillover” i.e., the general tendency to trust like-minded individuals, which may result in bias toward in-group politicians.
2. The in-group favoritism (linked to political intergroup relations) i.e., the tendency to trust in-group partisans or political leaders, which stems from moral values and social attitudes of voters.

Thus, the inserted segments provide clues on the relationship between the ensuing results from the economic trust game, and the results from PCA analysis on moral values and social attitudes of voters. The more elaborated insight on the nature of this relationship is detailed in the closing discussion of the paper.

IV. Adaptations in the MATERIALS AND METHODS

The explanations regarding the different periods/seasons for realization of the main/control experiment, are highlighted (in yellow) and copied below:

“Two experiments were conducted in two consecutive periods during 2014 (Exp 1: January-July and Exp 2: October-December) with the use of two famous characters in the role of trustee as a main difference. Specifically, the main experiment (Exp 1) utilized Silvio Berlusconi, former Prime Minister of Italy and leader of the center-right parties' coalition, while the control (Exp 2) utilized Piero Angela, a famous TV host. We decided to conduct our main experiment throughout one continual semester, specifically for the purpose of avoiding the confound of the season on the public perception of Berlusconi. Namely, we wanted to avoid the possibility that unexpected political developments might affect the public image of Berlusconi. Indeed, a political trust-index may quickly fluctuate due to the constant political engagements and public affairs of the concerned leader. However, no newsworthy political events happened throughout the period of Exp1. This was also evidenced in the index of trust towards Berlusconi which was monitored with bimonthly reports obtained by “Istituto Piepoli” (an agency specialized in marketing and opinion research) revealing 31%, 31%, and 34% of the total voting-pool, for the months of January, April, and June.” (page 4, lines 53-58)

“..In addition, we did not expect considerable fluctuation in the public perception of Angela, hence we were not restrained by the season for the Exp2. He is a famous non-politician with long-term media presence, but stable public appearance. This was confirmed in both, the pilot study and the experimental study, where participants on both sides did not differ in their perception of Angela and considered him as a relatively neutral character.” (page 5, lines 32-36)

The additional information is provided in concordance with the recommendations by Reviewer 1, and follows faithfully his/her instructions, which are cited in the following segment:

“Finally, I had not noticed the seasonal confound of the conditions in this experiment in my initial review, but I think it would be helpful to see more explanation about why running the two conditions in totally different seasons is not a worrisome confound. If participants had been randomly assigned to condition within the same season of data collection, the authors would still not have had the concern that perceptions of Berlusconi would be changing. I think it would be helpful to demonstrate that there were no newsworthy events in terms of politics or entertainment during or between the two seasons of data collection that could have affected participants’ perceptions.”

V. Adaptation in the RESULTS

The main intervention in this part of the manuscript, includes the title of the section about the Principal Component Analysis (PCA) from data, regarding the moral values and social attitudes of voters. It was formerly known as "Inferred ideology as indicator of economic behavior" while the updated version is highlighted below (in yellow) and now reads as follows: **“GROUP BINDING DIMENSION AS POSSIBLE INDICATOR OF ECONOMIC BEHAVIOR”** (page 7, line 26)

Consequently, we adapted segments belonging to this section which described the results from the PCA analysis i.e., the emerging set of factors (Group Binding Dimension and the Social Equality Dimension). The new passage now reads as follows:

“The first factor was named Group Binding Dimension, since it comprised binding moral values together with the RWA attitudes, thus tapping onto the social conformity construct. The second factor was named Social Equality Dimension, since it comprised the remaining moral values concerned with fairness and harm reduction (individualizing MFQ variables), decreased proneness to hierarchy (SDO) and legitimization of economic inequality (ESJ), thus tapping onto the social equality construct. Clearly, the first factor is associated with group-centrism and

possible group-related favoritism, so we decided to proceed with the investigation on the interaction between the degree of Group binding dimension and the economic behavior of voters indexed by the average amount of investment.” (page 7, lines 42-51)

The PCA was obtained from the voters' responses on Moral Foundation Questionnaire, Right-Wing Authoritarianism Questionnaire, Social Dominance Orientation Scale and Economic System Justification Questionnaire. As mentioned, it resulted in two emerging factors, which were originally named Social Conservatism and Economic Liberalism. However, in the revised version of the manuscript we decided to rename the factors into Group Binding and Social Equality Dimension, accordingly. We believe that the updated titles are more suitable for the following reasons:

- They are better coordinated with the rest of content and the presenting style of the revised manuscript. Specifically, the title of the first factor addresses directly the possible reasons for the demonstrated in-group bias (for the group of supporting voters towards their own leader) i.e., the group conformity.
- The updated names are more accurate and better corresponding with their constituting variables. Specifically, the first factor which is comprised of the so-called binding group of moral values (i.e., Group Loyalty, Respect for Authority, and Purity/Sanctity) and the Right-Wing Authoritarianism attitudes, is re-named in such a way, as to address the common denominators and the main hallmarks of the tendency to comply and conform with one's own group.
- Finally, the updated name of the first factor, is better coordinated with the existing literature, and continues the thread about the positive correlation between the authoritarianism attitudes and the endorsement of binding moral foundations (Kugler et al., 2014).

As for the technical aspects of the RESULTS section, we followed the suggestions by Reviewer 2 to **“specify clearly what the IV, DV, and covariates are to the reader”** and properly annotated the variables where applicable. Most importantly, we clarified that the Group Binding Dimension was used **“as a single-item measure”** and **“the IV in our analyses”**, as per suggestion from Reviewer 2.

V. Adaptations in the DISCUSSION

We hereby confirm that the Discussion underwent most extensive restructuring, in order to fit with the more rigorous style in presenting/interpreting facts and results, and thus complies with reviewers' suggestions.

Specifically, in the updated version of the Discussion we removed all arguments and theoretical concepts regarding about the so-called "cognitive rigidity" or "holistic cognition" which were formerly associated with the behavior of the RW-group toward their leader. Namely, they were neither directly tested nor manipulated in our study, and were linked only hypothetically with the interpretation of our results, so we decided to abstain from mentioning them.

This was done in accordance with the decision to provide more factual (rather than hypothetical) shaping of the manuscript, and in concordance with the following suggestion from Reviewer 2:

"It would be important for the authors to completely flesh out why adjustment/trustworthiness were considered to be proxies for the "cognitive rigidity" concept. Their theoretical links could benefit from guided clarity. I read too many names of concepts and mechanisms and did not see how those were manipulated or measured in the study, or how they were related to the relationship between ideology and trust. The mechanisms were technically not directly tested, but were rather hypothesized. Maybe further studies could elaborate on the mechanisms."

The removed arguments, were replaced with a more elaborated emphasis on our main results. In the following lines we will copy the excerpts which are self-explanatory and expand on the concept of the in-group political bias.

"Our central finding (related to the lack of adjustment in the group who played with their un/trustworthy political leader) provides evidence beyond the existing literature on the intergroup ideological bias (i.e., the automatic preference for the members of one's own political group), guided by the preference for one's own political leader. Also, it supports the so-called 'epistemic spillover', a flawed heuristic of trusting like-minded political figures in non-political matters (in this case, economic trust decisions in an interactive game). Moreover, our findings suggest the mechanisms that are at the core of ideological intergroup biases, by highlighting the role of two important dimensions, especially prominent in RW-voters: a) the strong group-

binding moral values and b) the respect for the in-group authority figures (i.e., powerful leaders).” (page 8, lines 27-35)

“These results are in line with the Social Identity Theory regarding expressive partisan identity [7], the Moral Foundation Theory (MFT) and the Right-Wing Authoritarianism Theory (RWA) [8, 9, 10, 24]. According to MFT, the five basic moral foundations (harm/care, fairness/reciprocity, in-group/loyalty, authority/respect and purity/sanctity), collapse into two super-ordinate foundations. Specifically, the first couple of values are labeled as individualizing foundations (generally oriented toward protection and fair treatment of individuals), while the remaining three as binding foundations (focused on protection of the group, collectives, institutions). The binding foundations include: a) patriotism and self-sacrifice for one’s group (in-group loyalty); b) concerns about the importance of social order, traditions and respect for leadership (authority/respect) and; c) prevalence of spiritual over the carnal nature of humans (purity/sanctity). The political identity largely shapes people’s moral foundations, with LW-voters endorsing the individualizing foundations, while RW-voters ascribing same or higher moral relevance to the binding foundations. The latter group of values have been examined in the context of immorality i.e., ‘unacceptable behavior, such as blind obedience and stigma’ [9].” (page 8, lines 49-58; page 9, lines 6-10)

“In an analogous fashion, voters can ‘use the perceived credibility of political figures as a heuristic to guide their evaluations’ of what is right or wrong, and decisions of whether to trust or not i.e., the effect known as ‘epistemic spillover’ [13, 14]. Accordingly, our previous studies have already shown that the perceived similarity between voters’ and leaders’ personality, can influence even the basic cognitive processes of voters, such as attentional gaze capture [4, 16, 28]. Our current experiment provides more direct evidence, showing that voters’ shared political ideology with an in-group political leader ‘spills over’ to their economic decision-making processes in a trust economic game (i.e., unrelated, non-political context).” (page 9, lines 34-42)

As a concluding remark, the ultimate reason for adding more elaborate explanations regarding the obtained results, is the striving toward improved understanding about in-group political bias, and the situations/circumstances under which it may appear.

VI. Adaptations in the REFERENCED LITERATURE

Along with the removal of the arguments on the concepts of “cognitive rigidity” or “holistic cognition” (formerly presented in association to the cognitive styles of conservative voters), their supporting literature was also removed from the reference list (as unrelated and irrelevant for the restructured version of the manuscript).

Instead, additional set of references were introduced regarding the flawed heuristics, and potential in-group political biases of voters. Specifically, the following references were added to the list:

[13] Marks J, Copland E, Loh E, Sunstein CR, Sharot T. 2018. Epistemic spillovers: Learning others' political views reduces the ability to assess and use their expertise in nonpolitical domains. *Cognition*. (doi: 10.1016/j.cognition.2018.10.003).

[14] Swire B, Berinsky AJ, Lewandowsky S, Ecker UK. 2017. Processing political misinformation: comprehending the trump phenomenon. *R. Soc. Open. Sci.* 4(3):160802. (doi: 10.1098/rsos.160802).

[34] Schepisi M, Panasiti MS, Porciello G, Bufalari I, Aglioti SM. 2019. Left threatened by right: political intergroup bias in the contemporary italian context. *Front. Psychol.* 10:26. (doi: 10.3389/fpsyg.2019.00026).

Appendix C

SAPIENZA
UNIVERSITÀ DI ROMA

RESPONSE TO THE REVIEWER

We thank the reviewer for the final round of comments with clear, concise and concrete messages. All along the revision process, they repeatedly proved to be very insightful and helpful in resolving all remaining dilemmas. In addition, we are grateful for the continual patience and diligence with the revision, and feel compelled to include a mention in the acknowledgment section. The shared efforts in our joint quest toward perfecting of the material were indeed essential in achieving what we believe is a well-balanced and well-rounded story.

We followed the recommendation of the RSOS Editors, and compiled our thoughtfully premeditated response, to be as specific as possible in addressing the minor remaining issues. We also revised the manuscript accordingly. We thus believe to have substantially improved the overall clarity of the manuscript, and it is our most sincere hope that the response will help in reaching a shared and satisfied regard on the material at hand.

Legend:

The reviewer's comments are cited in "quotes", marked ***with bold and italics, and highlighted in red.*** The revised segments of the manuscript are also are copied as cited in "quotes", **marked with bold, plus highlighted in yellow.** In the framework of this response, a copy of the updated manuscript is also provided (**highlighting the revised segments in yellow**).

Sincerely,

The authors

COMMENT No.1

(1) “The added reasoning about the season/condition confound in the experiment is helpful, but it does not fully reassure the reader that there is not a seasonal (perception) confound with the treatment conditions that could be potentially affecting participants’ behavior. I think the clearest way to show that the season of experimental administration did not affect the experiment would be to show (relative) invariability in public perceptions of both Berlusconi and Angela over the full span of the experiment, January to December (not just Berlusconi and not just January to June). If the authors have access to such perception polls, it would do a lot to address the confound.”

RESPONSE:

Once again, we are grateful for the continual striving toward clarification of all potentially unresolved dilemmas. More importantly, we thank the reviewer for suggesting concrete steps to resolve this specific matter. In the case of Berlusconi, we do have access to public perception polls for the mentioned period, and we include them in the revised manuscript in the following manner:

“We conducted our main experiment over the course of one semester, for the purpose of avoiding unexpected political developments which could affect the public image, and result with fluctuations in the public trust-index toward Berlusconi. In order to control for a potential seasonal confound, we also consulted public electoral and political polls (managed by the Presidency of Ministers and the Italian Department of Information and Publishing)¹. Only one agency (“IPR Marketing”) provided reports throughout the whole year (December 2013, May 2014, December 2014). According to these reports trust-index toward Berlusconi proved to be fairly consistent for the investigated period (25%, 23% and 20% respectively).” (page 5, lines 1-9)

As for the control character (i.e., Piero Angela), in the absence of similar public perception polls (since he is not a politician and public decision-maker), we resorted to the results from the pilot study (conducted at the start of the experimental period), and the experimental study (conducted all through the end of the experimental period), and reported them accordingly. The copied excerpt

1 Accessed through the website <http://sondaggipoliticoelettorali.it/>

refers only to the part of the manuscript which was revised in accordance with the suggestion by the reviewer. For comprehensive information regarding the performed analyses and the obtained results, please refer to the preceding/following segments which were already included in the manuscript.

“In addition, we did not expect considerable fluctuation in the public perception of Angela, because he is a famous non-politician with long-term media presence, but stable public appearance. Hence, we were not restricted by the period and conducted our Exp. 2 in the second half of 2014. However, in order to control for the potential seasonal confound, we relied on our studies (since there are no available public perception polls for Angela). Namely, in our pilot (conducted at the beginning of 2014) and the experimental study (conducted toward the end of 2014), the participants on both sides of the political spectrum, did not differ significantly in their perception of Angela and considered him as a relatively neutral character.” (page 5, lines 33-40)

Also, we performed an exploratory search through the scientific databases, on the potential influence of the season in the outcomes of an economic trust game. To the best of our knowledge (and given the restricted period), we could not detect any such study confirming (or even exploring) the relationship. However, we reserved the possibility for such explorations in the future, by including a mention in the discussion section.

“Future studies with longitudinal explorations (spanning across multiple seasons), and diverse geopolitical settings (especially countries with a strong left-wing equivalent), will shed important light on the subject.” (page 10, lines 39-42)

COMMENT No.2

(2) “I think the reported group binding model is not as informative as it could be. It seems that the purpose of including this analysis is to demonstrate that this group binding dimension helps to explain why RW participants might be exhibiting this trusting behavior of Berlusconi. But the included model simply shows that people who are higher in group binding psychological preferences also exhibit this trusting behavior. I think it would make more sense to include these

psychological variables with (rather than instead of) the ideology variable in the model to see if the psychological variables help to explain the relationship between ideology and behavior. Furthermore, it would be even more informative—given these variables were measured—to include not only the group binding measure but also the social equality measure in the model to test whether the group binding measure does a better job of explaining the variance than preferences for equality, as the authors may be suggesting. At minimum, it seems plausible that participants' economic preferences and their perceptions of Berlusconi's economic positions could play a role in their experimental behavior."

The analysis on the group binding dimension primarily serves to provide an explanation for the economic behavior of the groups of participants toward an in-group/out-group leader (as compared to a control). However, the participants were divided according to their ideology (in LW/RW group), so the results obtained from the analysis on the group-binding dimension, can also serve to provide an indirect (inferred or implied) insight, on the link between the political ideology and economic behavior, when relating to an in-group/out-group leader. In that sense, the reviewer's remark is indeed: a) very intuitive, as there was a statistically significant positive correlation ($r = 0.71$, $p < .001$) between the two variables (i.e., self-reported ideology and the group-binding dimension); and b) much in line with the existing literature showing that those two variables are indeed strongly related (Kugler, 2014). The significant positive correlation between the variables was the main reason why we decided not to include both in the same linear mixed effects model, since they break one of the underlying assumptions of non-collinearity and suffer from redundancy. We also updated the manuscript accordingly, by adding the correlation results as follows:

"Specifically, we decided to repeat the aforementioned LME with only one procedural modification: the self-reported political group in the full model i.e., Group (RW/LW) was substituted with the score on the Group binding dimension (extracted with PCA from the questionnaires and used as a single-item measure i.e., independent variable), since the correlation between the two variables was strong ($r = 0.71$, $p < .001$)." (page 8, lines 4-8)

As for the notion regarding the inclusion of the Social equality measure in the LME analysis, we hereby like to confirm that we did run the required analysis, and in accordance with our expectations, the results revealed that the full-interaction model was not marked by AIC drop, nor by significant change in the LR (as compared to its nested model). This served as an additional confirmation for the decision to focus on the group-binding dimension in our analysis, which indeed yielded additional insight on the economic behavior of the participants when playing with an in-group/out-group leader. However, to provide all the details related to our study, we have included the negative results of this analysis in the SI (pages 9-10).

COMMENT No. 3

(3) “A very minor point: How correlated was participants’ voting behavior in the past three elections with their explicit political orientation (p. 15)? It’s stated that this question was included to verify participants’ understanding of their own ideology, but how this information was used would be even more helpful.”

The reviewer is right to observe that an explanation regarding the past voting behavior, would increase the understanding on its particular role in the experiment. Therefore, we use the opportunity to clarify that these data were employed as an additional control upon selection of participants (i.e., an extra safety measure, since otherwise, the selection would rely solely on the self-reported political orientation). Namely, the main purpose of those control questions was to detect participants with conflicting responses (i.e., those who self-identified as belonging to one of the RW/LW groups, while stated to have voted for a party belonging to the other LW/RW group), and to exclude them from the experiment. A summary explanation is also added in the manuscript:

“The questions regarding the past voting behavior were introduced for the purpose of excluding participants who did not have a clear idea about their political orientation and gave conflicting responses (e.g., self-identified as pro-left, while reported voting for pro-right political parties, and vice versa).” (page 4, lines 14-17)

However, in order to directly address the question on the relationship between the self-identified political orientation on one hand, and the actual political choices of our participants on the other, we ran a rank-biserial correlation between the two variables. Participants' political orientation was measured via direct question i.e., by asking them to self-place on a 7-point Likert-type item (1="extremely left"; 2="left"; 3="center-left"; 4="center-right"; 5="right"; 6="extremely right", 7="apolitical"). Those choosing "7" were coded as "N/A". Participants' party preference at the recent elections was coded on a binomial scale and two groups were created: a group who voted for pro-RW parties (coded as "1") and a group of participants voting for pro-LW parties (coded as "0"), while those who chose "a big tent" party (like "Movimento 5 Stelle") or abstained from voting, were coded as "N/A". The results revealed a rank-biserial coefficient $r_{rb} = .95$ (Glass, 1965)². Hence, it is safe to conclude that subjective declaration of one's political orientation, and one's objective voting behavior proved to be mutually correlated. In this way participants' self-assignment to one of the groups (LW/RW) was verified.

2 Glass, G. V. (1965). A ranking variable analogue of biserial correlation: Implications for short-cut item analysis. *Journal of Educational Measurement*, 2(1), 91-95.

COMMENT No. 4

(4) “Finally, as I mentioned in my initial review, it would be helpful to include some discussion about the political implications of these results. That is, how does this simulated one-on-one trust game exchange with a famous politician (which is unlikely to happen in the real world) reflect realistic voter behavior and experience? It would be really helpful to make the potential links between the experimental context and actual political outcomes more explicit.”

We agree with the perceptive remark that the voter-leader interactions in the real world (as opposed to those in the experimental context) may seem more distant, less tangible and foreseeable, if not properly highlighted (through potential links). In fact, we believe that this is an excellent observation as it provided a rationale for our key opening point of the discussion. Therefore, we included the possible analogies and implications of the tested voter-leader interaction, in the introductory part of the discussion which now opens as follows:

“The trust-game dynamics between the political leaders and their devoted followers, occurs on a daily basis. The leaders usually assume a role of trustworthy subjects, by offering economic incentives (investments, credits and other financial benefits) to their trusting followers. In addition, the trust-game dynamics is also reflected in the fact that each voter is also a tax-payer (i.e., trustor), while the political leaders with governmental experience are decision-makers (i.e., trustees). They can choose to spend the budget for societal improvements (thus returning the trust and behaving trustworthy), or for their own benefits (thus behaving in an untrustworthy manner). Our approach allowed us to shed light on important aspects of these voter-leader interactions.” (page 8, lines 18-

25)

Subject Category:

Psychology and cognitive
neuroscience

Subject Areas:

group behavior, human
decision-making, trust

[revised manuscript text omitted]

Permission to carry out fieldwork: No special permissions were required prior to the conducting of the research.

Funding: The research was supported by the grants of national research interest (PRIN: Progetti di Ricerca di Rilevante Interesse Nazionale, Edit. 2015, Prot. 20159CZFJK and Edit. 2017, prot. 2017N7WCLP), as well as the ERC Advanced Grant, 2017 (eHONESTY, 789058).

Acknowledgments: We are grateful to Maria Serena Panasiti for helpful suggestions, Adriano Acciarino and Gianluca Angeli for their help in data collection. **Also, we want to express our gratitude to the anonymous reviewers for their diligent revision and substantial contribution in the improvement of the manuscript.**

[revised manuscript text omitted]

37 Schilke O, Reimann M, Cook KS. 2015 Power decreases trust in social exchange. *Proc. Natl. Acad. Sci. U.S.A.* 112(42):12950–12955. (doi: 10.1073/pnas.1517057112).